# ENFORCE: Nonlinear Constrained Learning with Adaptive-depth Neural Projection

## Abstract

Ensuring neural networks adhere to domain-specific constraints is crucial for addressing safety and ethical concerns while also enhancing inference accuracy. Despite the nonlinear nature of most real-world tasks, existing methods are predominantly limited to affine or convex constraints. We introduce ENFORCE, a neural network architecture that uses an adaptive projection module (AdaNP) to enforce nonlinear equality constraints in the predictions. We mathematically prove that our projection mapping is 1-Lipschitz under mild assumptions, making it well-suited for stable training. We evaluate ENFORCE on multiple tasks, including function fitting, a real-world engineering simulation, and learning optimization problems. For the latter, we introduce a class of scalable optimization problems as a benchmark for nonlinear constrained learning. The predictions of our new architecture satisfy $N_C$ equality constraints that are nonlinear in both the inputs and outputs of the neural network, while maintaining scalability with a tractable computational complexity of $\mathcal{O}(N_C^3)$ at training and inference time.

## 1 Introduction

Neural networks (NNs) are the backbone of many recent advancements in artificial intelligence (AI), excelling in tasks such as natural language processing, image analysis, and scientific discovery due to their modularity, simplicity, and strong generalization capabilities. However, their ability falls short when strict adherence to domain-specific constraints is required. Depending on the task, prior knowledge about the system (e.g., from physics, safety, or ethics) is often available and is typically leveraged by humans in decision-making processes. In contrast, data-driven methods such as NNs rely solely on data. Thus, trained NNs may be accurate on a training and test data set but still may not satisfy known constraints, leading to inconsistent predictions. This limitation not only generates substantial skepticism, hindering their adoption in real-world applications, but can also lead to erroneous or physically infeasible outcomes in decision-making processes. Moreover, when domain knowledge is available as analytical equations, ensuring that NNs adhere to this information is crucial to avoid a suboptimal utilization of expert insights and potentially reduce data demand (E. Samadi et al., 2022).

Enforcing strict constraints in NNs is a promising area of research for many fields. For example, in AI for Science, integrating first-principle laws ensures physically consistent models, enabling insightful scientific discovery (Wang et al., 2023; Xu et al., 2021) or system modeling in engineering (Schweidtmann et al., 2024). A prominent area of application involves using NNs to safely accelerate computationally intensive tasks such as learning surrogate models (Lastrucci et al., 2025) or solutions to (parametric) constrained optimization problems (also known as *proxy optimization*), either in a supervised or unsupervised manner (Kotary et al., 2021; Donti et al., 2021; Di Vito et al., 2024; Schweidtmann & Mitsos, 2018). More broadly, constraining neural network predictions can have a transformative impact in domains where strict adherence to critical requirements is essential, including safety-critical systems (Gupta et al., 2021; Gerke et al., 2020), bias mitigation (Feuerriegel et al., 2020; Hardt et al., 2016), and compliance with regulatory standards (Cao, 2022). Additionally, with the rise of generative AI (GenAI), enforcing constraints on generation processes could mitigate risks by ensuring that generated data respects given criteria (Li et al., 2025). Constraining the neural network output to adhere to strict rules is also beneficial to tackle traditional machine learning challenges, such as overfitting in data-scarce regimes (Min et al., 2024).

Enforcing constraints in NNs is not straightforward. The majority of existing approaches rely

on incorporating penalty terms into the loss function to minimize constraint violations (Raissi et al., 2019). Yet, these penalty-based methods offer no guarantees of constraint satisfaction (*soft-constrained*). In contrast, other methods aim to ensure strict adherence to analytical constraints by design (*hard-constrained*). For instance, one can use sigmoid functions to bound outputs. To enforce analytical constraints, recent studies incorporate correction layers into NNs to project or complete the predictions, ensuring they lie within a feasible region. For example, developed methods can enforce constraints defined by affine relationships between input and output variables or by convex regions (Chen et al., 2024; 2021; Min et al., 2024; Iftakher et al., 2025). However, many applications, e.g., in science or sociology, are inherently governed by nonlinear constraints (Mize, 2019; Nicolis, 1995). Existing approaches for handling nonlinear constraints predominantly rely on external root-finding or constrained optimization solvers (Donti et al., 2021; Mukherjee & Bhattacharyya, 2024; Iftakher et al., 2025). These methods introduce significant computational overhead and complicate model development, thereby compromising the modularity and flexibility typically associated with NNs.

We propose ENFORCE, a neural network architecture that enforces predictions to satisfy nonlinear equality constraints. ENFORCE is trained using standard unconstrained optimization techniques and leverages an adaptive-depth neural projection (AdaNP) module to enforce constraints by construction without relying on external solvers. We evaluate ENFORCE on multiple problems, including a real-world engineering simulation and a scalable class of nonlinear parametric optimization problems that we propose as a benchmark for nonlinear constrained learning.

## 2 RELATED WORK

This section reviews different approaches to enforcing constraints in NNs, with a focus on existing hard-constrained methods.

### 2.1 SOFT-CONSTRAINED NEURAL NETWORKS

One of the earliest approaches to embedding domain knowledge into NNs involves the use of *soft* constraints. Soft constraints are incorporated as penalty terms appended to the loss function, penalizing residuals of algebraic (Erichson et al., 2019; Pfrommer et al., 2020) or differential equations underlying the system (Wang et al., 2021). Physics-informed Neural Networks (PINNs) (Raissi et al., 2019) represent a widely used framework designed to solve partial differential equations (PDEs) with deep learning by employing soft constraints and collocation points. Although the soft-constrained approach places no restrictions on the complexity of the constraints, it has the drawback of not guaranteeing strict adherence to them. Furthermore, the complication of the loss landscape – especially when the different terms vary in nature or scale – can degrade the optimization performance of the neural network, often resulting in suboptimal accuracy (Wang et al., 2020a;b).

### 2.2 HARD-CONSTRAINED NEURAL NETWORKS

Hard-constrained neural networks refer to methodological approaches ensuring that neural network predictions adhere to analytical constraints by construction. These constraints, explicitly encoded within the architecture, act as inductive biases, guiding the learning process toward compliance with domain knowledge or restrictions (Karniadakis et al., 2021). Architectures such as convolutional neural networks (CNNs) (LeCun et al., 1989) and graph neural networks (GNNs) (Bronstein et al., 2017; Wu et al., 2021) encode inductive biases by guaranteeing invariance with respect to patterns and symmetries. Simple analytical constraints can be enforced using differentiable functions, such as sigmoids or ReLU for output bounding and softmax for simplex constraints. Recent literature includes significant contributions for enforcing analytical inequality constraints, such as convex polytopes and convex sets more generally (Frerix et al., 2020; Donti et al., 2021; Wang et al., 2024; Tordesillas et al., 2023; Konstantinov & Utkin, 2023). One can also constrain the neural network to guarantee specific functional characteristics, such as Lipschitz continuity (Anil et al., 2018) or Lyapunov stability (Manek & Kolter, 2020). Nevertheless, this falls outside the scope of this study. For a broad and recent review on hard-constrained NNs, the reader is also referred to (Min et al., 2024). Since this paper focuses on analytical equality constraints, the following literature review considers existing methods for this specific case.

**Projection methods**   Many methods for encoding hard equality constraints utilize projection techniques, which correct preliminary neural network predictions by appending a non-trainable layer to the output. Projections can be formulated as optimization problems (i.e., distance minimization) or derived from geometric principles. For example, in Chen et al. (2021) neural network predictions of physical systems governed by PDEs are projected to ensure solutions satisfy the finite difference discretization of the underlying linear PDEs. A more general approach is the KKT-hPINN, which enforces linear equality constraints in the inputs and outputs (Chen et al., 2024). Recently, Hard-Net was introduced to enforce equality and inequality constraints affine in the output, without input restrictions, via a closed-form projection step (Min et al., 2024). Moreover, Iftakher et al. (2025) proposed a method to enforce nonlinear constraints leveraging log-exponential reformulation and a Newton method.

**Predict-and-complete**   NNs can also predict a subset of output variables, $y_P \in \mathbb{R}^{N_O - N_C}$, and complete the prediction by solving the system of constraints based on this partial output (null-space methods). This approach ensures that the constraints are always satisfied. For instance, Beucler et al. (2019) introduced this concept to simulate physical systems such as climate modeling. However, when the constraints are not available in explicit form, solving the system requires a root-finding solver. Similar approaches have been proposed within the hybrid modeling community, particularly in the *serial* configuration, where a fully data-driven method is used to predict unknown inputs to a mechanistic model (Schweidtmann et al., 2024). While studies like DC3 (Donti et al., 2021) have developed efficient backpropagation techniques, scenarios involving implicit nonlinear constraints can be computationally expensive to tackle with predict-and-complete methods. Moreover, we rigorously show in Appendix B.6 that predict-and-complete approaches can suffer training instabilities if the constraints Jacobian is ill-conditioned (Beucler et al., 2019).

**Constrained optimization**   To enforce analytical constraints, researchers leveraged constrained optimization to deploy specialized layers or directly train the neural network. OptNet (Amos & Kolter, 2017) is an optimization layer developed to solve quadratic programs. Agrawal et al. (2019) expand the methodology to convex programs. They develop efficient differentiation techniques through such layers. Min et al. (2024) leveraged such optimization layers to develop HardNet-Cvx, a neural network enforcing convex constraints. However, the forward pass always requires the solution of a constrained optimization problem. Recently, Mukherjee and Bhattacharyya (Mukherjee & Bhattacharyya, 2024) approached the constrained learning paradigm by training a neural network using a constrained optimization solver such as IPOPT (Wächter & Biegler, 2005) instead of standard unconstrained optimization algorithms. However, these approaches pose severe limitations in terms of NNs and dataset size.

**Other methods**   Other methods have been proposed for constrained learning in NNs, mostly considering affine or convex regions (Tao et al., 2023; Tao & Thakur, 2024). Many of them consider constraints only dependent on the input of the neural network (Schweidtmann et al., 2021; Tordesillas et al., 2023; Balestriero & LeCun, 2022; Brosowsky et al., 2020), others design strategies to include the dependence on both inputs and outputs (Konstantinov & Utkin, 2023; Lastrucci et al., 2025). Recently, contributions to enforce general logic and linear constraints have been proposed by the neuro-symbolic AI community, developing loss terms or constraining layers using logic programming (Giunchiglia & Lukasiewicz, 2021; Stoian et al., 2024; Fischer et al., 2019). However, to the best of our knowledge, no existing method enforces nonlinear equality constraints involving both the input and output of a neural network by embedding them into the architecture while allowing training with unconstrained solvers such as Adam (Kingma & Ba, 2014) or relying on Newton solvers.

## 3   PRELIMINARIES

**Problem statement**   Given a dataset $(x_i^*, y_i^*)_{i=1,...,N}$, without loss of generality we consider a neural network $f_\theta$ with parameters $\theta$ to approximate the underlying relationships while satisfying a set of known algebraic equality constraints $c(x, y) = 0$. In general, $c$ can be a nonlinear function in the input $x$ and output $y$ of the neural network, incorporating domain knowledge or specifying critical requirements. Similarly, $f_\theta$ can be any neural network architecture.

**Assumption 1.** *i) The constraints $c(x, y) = 0$ are feasible and linearly independent, ii) $N_C < N_O$, where $N_C$ is the number of equality constraints and $N_O$ is the output dimensionality of the neural network, i.e., there are available degrees of freedom to learn.*

One way to enforce the neural network prediction $\hat{y}$ to satisfy the constraints is to project it onto the feasible hypersurface (manifold) defined by $c(x, y) = 0$. The projection operation can be defined as an optimization problem:

$$\tilde{y} = \arg \min_y \frac{1}{2} (y - \hat{y})^T W (y - \hat{y}) \quad \text{s.t.} \quad c(x, y) = 0 \tag{1}$$

If $W$ is the identity matrix, the prediction is corrected by an orthogonal projection onto the feasible region. This can be interpreted as finding the feasible solution that minimizes the Euclidean distance from the original prediction $\hat{y}$. A local solution to the nonlinear program in Eq. 1 can be found by solving the first-order necessary optimality conditions, known as Karush–Kuhn–Tucker (KKT) conditions (Nocedal & Wright, 2006). However, the latter is not necessarily straightforward as it may involve solving a system of nonlinear equations.

**Quadratic projection** When $c(x, y)$ is an affine function in the neural network input and output, then the problem results in a quadratic program (QP) and a closed-form analytical solution is available for the KKT conditions (Chen et al., 2024). An extension to the closed-form is available when generalizing to any function in the input $x$, as the projection operation is still a QP. Consider an affine constraint on $y$ of the form $c = C(x)y - v(x) - b = 0$, where $C(x)$ and $v(x)$ act as the linear coefficient matrix and translation vector, respectively. For a given input $x_i$ and prediction $\hat{y}_i$, any function of $x_i$ can be treated as constant with respect to the optimization problem in Eq. 1, which thus reduces to a QP.
Enforcing affine functions is not new and is also achieved through other techniques (Min et al., 2024; Balestriero & LeCun, 2022). However, relaxing the assumption to allow for nonlinear constraints of both the input and output of the neural network commonly results in decreased computational efficiency and stability, as it typically requires the use of constrained optimization (Mukherjee & Bhattacharyya, 2024) or root-finding solvers such as Newton's methods (Donti et al., 2021; Iftakher et al., 2025).

## 4 NONLINEAR CONSTRAINED LEARNING

We present ENFORCE, a framework designed for general and efficient nonlinear constrained learning. The method employs a computationally cheap adaptive neural projection module and has no restriction on the nonlinearity of $\mathcal{C}^1$ constraints involving both the input and output of the neural network. We prove the neural projection to be a 1-Lipschitz mapping, implying adversarial robustness and stable gradient flow dynamics when compared to state-of-the-art constrained learning methods such as predict-and-complete.

### 4.1 ADANP: ADAPTIVE-DEPTH NEURAL PROJECTION

We locally approximate the nonlinear program in Eq. 1 and exploit the efficiency of quadratic projections to generalize the methodology to any nonlinear constraint. Assuming $c$ is of class $\mathcal{C}^1$, we use first-order Taylor expansion to locally linearize the constraints around the neural network input $x_0$ and prediction $\hat{y}$:

$$c(x, y) \simeq c(x_0, \hat{y}) + J_x c|_{x_0, \hat{y}} (x - x_0) + J_y c|_{x_0, \hat{y}} (y - \hat{y}), \tag{2}$$

where $J_x c$ and $J_y c$ are the Jacobian matrices with respect to the variable $x$ and $y$, respectively. Since the neural network input is fixed for a given sample, the linearization is exact in $x$, thus, $x = x_0$. Considering orthogonal projection for notation simplicity, the nonlinear optimization problem in Eq. 1 is locally approximated by a (linearly constrained) QP:

$$\tilde{y} = \arg\min_y \frac{1}{2}\|y - \hat{y}\|^2 \quad \text{s.t.} \quad c(x,\hat{y}) + J_y c|_{x,\hat{y}}\,(y - \hat{y}) = 0 \tag{3}$$

**Definition 1** (Projection operator $\mathcal{P}$). *Given an input $x \in \mathbb{R}^{N_I}$ to a neural network $f_\theta$, its prediction $\hat{y} = f_\theta(x) \in \mathbb{R}^{N_O}$, and a set of constraints $c \in \mathcal{C}^1(\Omega, \mathbb{R}^{N_C})$, with $N_C < N_O$, we define an operator $\mathcal{P}$ such that $\tilde{y} = \mathcal{P}(\hat{y})$ is the solution to the linearized quadratic program in Eq. 3, in the domain $\Omega$ where the constraints are defined.*
*In particular, $\tilde{y} = B^*\hat{y} + v^*$, with $B^* = I - B^T(BB^T)^{-1}B$ and $v^* = B^T(BB^T)^{-1}v$, where $I \in \mathbb{R}^{N_O \times N_O}$ is the identity matrix, $B = J_y c|_{x,\hat{y}}$, and $v = J_y c|_{x,\hat{y}}\,\hat{y} - c(x,\hat{y})$.*

Given the closed-form expression of the operator $\mathcal{P}$ derived in Appendix B.1, we can define a differentiable *neural projection* (NP) layer representing the operator $\mathcal{P}$. The forward and backward passes of an NP layer are computationally cheap (more details on implementation and computational cost are given in Appendix C). However, the operator $\mathcal{P}$ projects the neural network prediction onto a linear approximation of the nonlinear constraints (i.e., the tangent hyperplane). The error that we introduce is proportional to the projection displacement $e_D = \|\tilde{y} - \hat{y}\|$. From this consideration, it follows that (1) the error is mitigated as the projection displacement is small, i.e., the neural network prediction is sufficiently accurate, and (2) a single NP layer cannot ensure exact adherence to nonlinear constraints. It is worth noting that a single NP layer guarantees strict satisfaction of equality constraints that are affine in $y$ and nonlinear in $x$, i.e., it efficiently enforces constraint classes considered in similar recent works (Chen et al., 2024; Min et al., 2024).

To address the challenge of satisfying nonlinear constraints, we propose AdaNP: an adaptive-depth neural projection composition that, under certain conditions, enforces nonlinear constraint satisfaction to arbitrary tolerance $\epsilon$.

**Definition 2** (AdaNP module). *Given an operator $\mathcal{P}$ as defined in Def. 1, AdaNP is a composition of $n$ operators $\mathcal{P}$, such that:*

$$AdaNP = \mathcal{P}_1 \circ \cdots \circ \mathcal{P}_n$$

**Proposition 1** (Convergence of AdaNP). *Given an arbitrarily small scalar $\epsilon$, $n \in \mathbb{N}$ and assuming $\hat{y}$ in the positive reach (cf. Def. 4, Appendix B.4) of the constraints manifold $\mathcal{M} = \{x \in \mathbb{R}^{N_I},\, y \in \mathbb{R}^{N_O} : c(x,y) = 0\}$, then $\tilde{y}_n$ is computed as:*

$$\tilde{y}_n = (\mathcal{P}_1 \circ \cdots \circ \mathcal{P}_n)(\hat{y})$$

*and converges to a feasible prediction such that $|c(x,\tilde{y}_n)| < \epsilon$ with linear convergence rate under constraint smoothness conditions (cf. proof in Appendix B.3).*

AdaNP is a differentiable stack of $n$-NP layers that can be composed on every neural network backbone. The depth $n$ adjusts adaptively during training and inference depending on the nonlinearities and the specified tolerance (cf. Algorithm 1 in Appendix C.1 for details about the adaptive behavior). Accurate NNs typically result in shallower AdaNP modules, since the linearization error $e_D$ is related to the distance $\|\hat{y} - y^*\|$ between the neural network prediction $\hat{y}$ and ground truth output $y^*$. This introduces a trade-off between the complexity of the backbone and the required depth of AdaNP to satisfy the specified tolerance criteria.

**Analogy with Sequential Quadratic Programming** AdaNP can also be seen as an iterative method that recursively improves the solution of a linearized nonlinear program. Here, we notice the similarity to sequential quadratic programming (SQP) techniques. Specifically, AdaNP is a simple case of SQP method for which the objective function is naturally quadratic while the nonlinear constraints are linearized (in contrast to full SQP, in which the objective function is quadratically approximated). This observation allows to analyze the convergence rate of the method starting from SQP theory (Nocedal & Wright, 2006; Fletcher & Leyffer, 2002; Fletcher et al., 2002). The reader is referred to the Appendix B.3 for a complete discussion.

**Deviation from Newton's method** While the KKT conditions for a nonlinear program (Eq. 1) can be more generally solved using Newton's methods, our method circumvents the computational overhead associated with calculating the Hessian matrix of the constraints (cf. Appendix B.2) at the cost of reduced convergence rate (i.e., full Newton's method converges quadratically).

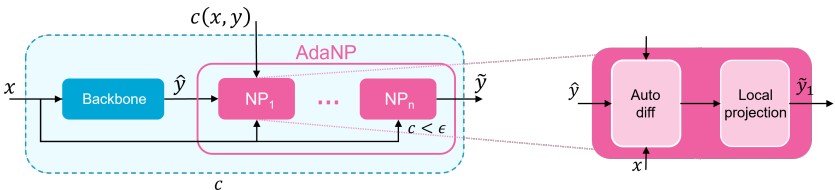

Figure 1: ENFORCE consists of a backbone neural network and an adaptive neural projection (AdaNP) module. The backbone network can be of every kind, such as fully connected, convolutional, or transformer architecture. AdaNP includes an adaptive number of neural projection (NP) layers, each composed of an auto-differentiation and a local projection step.

### 4.2 CONDITIONING ANALYSIS

The set of constraints $c(x, y) = 0$ describes an infinite-wide feasible region (i.e., a hypersurface) where the constraints are defined. Hence, one could ask whether the projections are unique as well as whether the projection mapping is stable and well-conditioned, and thus suitable for guiding the learning process. In Appendix B.4 and B.5 we provide rigorous theoretical conditions for the uniqueness and *no-worse* property of the projected prediction, along with evidence of regularity and robustness of our method, and show that other state-of-the-art constrained learning approaches do not guarantee these properties.

Regularity and robustness of appended layers (e.g., projection or null-space completion) can influence the stability of the neural network training. We prove the projection operation to be a *non-expansive* mapping in the neighborhood of the constraint manifold (i.e., its Lipschitz constant $L \leq 1$) under mild assumptions. This ensures adversarial robustness of the neural projection layer during the forward pass and stable gradient flow dynamics. Proofs and extensive discussion are given in Appendix B.6.

### 4.3 ARCHITECTURE

The architecture of ENFORCE (Fig. 1) is composed of (1) a neural network (without loss of generality) as backbone, which can be of any kind and complexity, and (2) an AdaNP module. The depth of AdaNP depends on the backbone performance and specified tolerance. Indeed, the tolerance of AdaNP can be tuned to increase training efficiency (Section 4.4). A single NP layer is composed of two steps: (1) automatic differentiation and (2) local neural projection.

**Exact Jacobian computation** To compute the Jacobian of the constraint system, when not available analytically, we leverage automatic differentiation available in most deep learning libraries (Paszke et al., 2019; Abadi et al., 2015; Bradbury et al., 2018). Computing the local Jacobian $J_y c|_{x,\hat{y}}$ is computationally inexpensive, as it requires propagating derivatives only through the constraints and does not involve the neural network backbone. Furthermore, its computation can be efficiently parallelized on GPU.

**Local neural projection** The neural projection defined by the operator $\mathcal{P}$ in Def. 1 depends on individual input-prediction instances. Thus, the projection is locally defined in the neighborhood of $(x_i, \hat{y}_i)$. We parallelize the computation of local neural projections by building a *rank-3* tensor $\mathbf{B}$ and a *rank-2* tensor $\mathbf{v}$ (Appendix C.3). Thus, we reduce the apparent complexity of an NP layer from $\mathcal{O}(BS \times N_C^3)$ to $\mathcal{O}(N_C^3)$, allowing effective training with stochastic gradient descent techniques with virtually no limitation on the batch dimension ($BS$). On modern hardware, handling up to $N_C < 10^3$ constraints results in a computational cost that remains practical, particularly when using Cholesky decomposition for matrix inversion (Burden & Faires, 2005). Moreover, the complexity of this method is *equivalent* to other state-of-the-art methods such as DC3 (Donti et al., 2021) and KKT-Hardnet (Iftakher et al., 2025). Additional insights into memory requirements are provided in Appendix C.4.

## 4.4 TRAINING ENFORCE

We train ENFORCE using standard unconstrained gradient descent methods (i.e., Adam). We develop and use a constrained learning methodology using AdaNP to guide the neural network training to convergence, supported by the theoretical implications of orthogonal projections described in Section 4.2.

**Loss function**  The loss function used throughout this study takes the following general form:

$$\ell = \ell_T + \ell_D + \ell_C = \ell_T + \frac{\lambda_D}{N} \sum_{i=1}^{N} ||\hat{y}_i - \tilde{y}_i||^2 + \frac{\lambda_C}{N} \sum_{i=1}^{N} ||c(x_i, \tilde{y}_i)||, \tag{4}$$

where the first term, $\ell_T$, is a task-specific loss function selected based on the target model. The second and third terms are regularization penalties that respectively minimize the projection displacement, $||\hat{y}_i - \tilde{y}_i||$, and the constraint residual $||c(x_i, \tilde{y}_i)||$. The relative contributions of these terms are controlled by the scalar weights $\lambda_D$ and $\lambda_C$. Minimizing the projection displacement aims to (1) ensure minor linearization error ($\epsilon_L \sim \Delta y$) and (2) prevent the neural network from learning alternative functions whose projections onto the constraints fall within the neighborhood of the desired functions. Also, this additional loss term is suggested to reduce reliance on AdaNP, thereby lowering the computational cost during inference (i.e., by decreasing the depth of AdaNP).

**Adaptive training strategy**  We propose a strategy to facilitate constrained learning during the early stages of training, guided by the theoretical insights presented in Section 4.2. In the initial training phases, the preliminary prediction $\hat{y}$ may be inaccurate and lie far from the constraint manifold $\mathcal{M}$. Under such conditions, projecting onto a locally linearized approximation of the constraints can introduce substantial errors in the prediction. To mitigate this issue in practice, inspired by trust-region methods (Nocedal & Wright, 2006), we activate AdaNP only when the projection operation leads to an improvement in the prediction accuracy (e.g., quantified by a decrease in some loss measure $m_\ell$). This often leads to an unconstrained pre-training phase, followed by the activation of the AdaNP module. In other words, this serves as a heuristic to ensure that the prediction $\hat{y}$ lies sufficiently close to the constraint manifold $\mathcal{M}$. Details about the algorithm and loss measures used in this study are reported in Appendix C.2.

## 5 EXPERIMENTS AND DISCUSSION

We evaluate the proposed method on different tasks: (i) learning solutions to scalable nonlinear optimization problems and (ii) a real-world engineering simulation on a chemical process. We also perform an extensive analysis of hyperparameters and training dynamics using an illustrative curve-fitting task, and we report additional examples with special constraints (see Appendix D). All experiments were conducted using an NVIDIA A100 Tensor Core GPU 80 GB, while the nonlinear programming solver (Section 5.1) runs on a CPU (11th Gen Intel(R) Core(TM) i7, 4 Core(s), 8 Logical Processor(s)).

### 5.1 CONSTRAINED OPTIMIZATION PROBLEM

A relevant field in NNs research involves learning approximate solutions to constrained optimization problems as an inexpensive alternative to traditional solvers. Existing benchmarks for such a task lack scalable problems involving nonlinear equality constraints, which limits the evaluation of methods beyond the linear setting. We address this gap by introducing a new benchmark that incorporates nonlinear equality constraints while retaining scalability in problem complexity, following a state-of-the-art protocol.
We compare our method with alternative baselines for learning (or solving) constrained optimization problems, such as MLP, soft-constrained MLP, the state-of-the-art DC3 (Donti et al., 2021), and the deterministic nonlinear programming solver IPOPT (Wächter & Biegler, 2005). We do not compare with baselines specialized for affine or convex constraints, such as RAYEN (Tordesillas et al., 2023) and HardNet (Min et al., 2024), as they are not designed to handle nonlinear equality constraints.

For every model, we use an equivalent fully-connected neural network backbone consisting of 2 hidden layers with 200 neurons. Training is performed with a batch size of 200, a learning rate of $10^{-4}$, and until model convergence (3,500 epochs for DC3; 1,000 for ENFORCE and the other baselines). Every run is repeated 5 times. We run the test inference on a single batch of 833 samples, and for the inference time of the optimizer, we assume full parallelization on 833 CPUs.

### 5.1.1 NONCONVEX PROBLEM WITH LINEAR CONSTRAINTS

We consider the same class of nonconvex optimization problems as in Donti et al. (2021), with the focus on equality constraints:

$$\min_{y \in \mathbb{R}^{N_O}} f_{\text{obj}}(y) = \frac{1}{2} y^T Q y + p^T \sin y, \quad \text{s.t.} \quad c^T y = x, \tag{5}$$

where $Q \in \mathbb{R}^{N_O \times N_O} \succeq 0, p \in \mathbb{R}^{N_O}$, and $c \in \mathbb{R}^{N_C \times N_O}$ are randomly sampled constant parameters, while $x \in \mathbb{R}^{N_C}$ (with $N_C = N_I$) is the variable parameter across problem instances. $Q$ is a diagonal matrix chosen to be positive semi-definite and $x$ is uniformly sampled in the interval $[-5, 5]$. We aim to learn the optimal $y$ given an instance of $x$ in an unsupervised fashion. Rather than using a dataset of solved optimization instances, we minimize the objective in the unsupervised task loss $\ell_T = f_{\text{obj}}(y) = \frac{1}{2} y^T Q y + p^T \sin y$.

Table 1: Results on a batch of 833 instances of nonconvex optimization problems with linear equality constraints involving 200 variables and 150 equality constraints. ENFORCE guarantees the feasibility of the solutions, is $25\times$ faster than IPOPT, and learns a $40\%$ better optimum than the state-of-the-art DC3 method. Baseline deep learning and soft-constrained methods show significant constraint violations and suboptimal predictions.

| Method | Obj. value | Max eq. | Mean eq. | Inference [s] | Training [min] |
|---|---|---|---|---|---|
| IPOPT | $-10.64 \pm 0.00$ | $0.00 \pm 0.00$ | $0.00 \pm 0.00$ | $0.379 \pm 0.060$ | – |
| MLP | $-52.99 \pm 0.01$ | $45.38 \pm 0.56$ | $9.14 \pm 0.02$ | $0.001 \pm 0.001$ | $9.0 \pm 0.2$ |
| Soft ($\lambda_c = 1$) | $-8.18 \pm 0.18$ | $1.47 \pm 0.41$ | $0.09 \pm 0.00$ | $0.001 \pm 0.001$ | $10.9 \pm 0.5$ |
| DC3 | $-6.27 \pm 0.07$ | $0.00 \pm 0.00$ | $0.00 \pm 0.00$ | $0.004 \pm 0.000$ | $25.2 \pm 8.6$ |
| **ENFORCE** | $-10.59 \pm 0.00$ | $0.00 \pm 0.00$ | $0.00 \pm 0.00$ | $0.016 \pm 0.002$ | $13.9 \pm 0.1$ |

In Table 1, we report the results on the constrained nonconvex task for 200 variables and 150 linear equality constraints. In Appendix D (Table 5), we show how the performance of the methods scales with varying numbers of variables and constraints. Given the linear nature of the constraints, ENFORCE consistently guarantees the feasibility for all the test samples. ENFORCE learns a solution that is only $0.47\%$ suboptimal relative to the objective value obtained by IPOPT, while obtaining a $25\times$ acceleration. ENFORCE shows faster training convergence when compared to DC3 and an optimal objective gain of $40\%$. This improvement can be attributed to the stability of the projection mapping, in contrast to the null-space completion method. As expected, unconstrained and soft-constrained methods do not guarantee feasibility and may yield predicted optima with lower objective values than those computed by constraint-respecting solvers. However, in constrained optimization, such infeasible solutions are inadmissible, regardless of their objective value.

### 5.1.2 NONCONVEX PROBLEM WITH NONLINEAR CONSTRAINTS

Extending the linear benchmark setting above, we introduce scalable problems with nonlinear equality constraints in both inputs and outputs, enabling systematic high-dimensional analysis beyond the linear case:

$$\min_{y \in \mathbb{R}^{N_O}} f_{\text{obj}}(y) = \frac{1}{2} y^T Q y + p^T \sin y, \quad \text{s.t.} \quad y^T A y + c^T y + d = x^3. \tag{6}$$

Here, $A \in \mathbb{R}^{N_C \times N_O \times N_O}$ denotes a tensor holding $N_C$ randomly generated symmetric matrices, while the remaining parameters follow the same sampling procedure as previously described. The varying parameter $x$ is uniformly drawn from the range $[-5, 5]$, and the dataset dimensionality and

Table 2: Performance comparison on nonconvex problems with nonlinear equality constraints involving 200 variables and 150 constraints. ENFORCE predicts feasible and near-optimal solutions across the entire test set, achieving a $25\times$ speedup over IPOPT. Other deep learning methods exhibit significant constraint violations and suboptimal performance.

| Method | Obj. value | Max eq. | Mean eq. | Inference [s] | Training [min] |
|---|---|---|---|---|---|
| IPOPT | $-29.45 \pm 0.0$ | $0.0 \pm 0.0$ | $0.0 \pm 0.0$ | $3.40 \pm 1.40$ | – |
| MLP | $-53.07 \pm 0.00$ | $497.4 \pm 4.6$ | $118.4 \pm 0.1$ | $0.002 \pm 0.001$ | $10.6 \pm 0.3$ |
| Soft ($\lambda_c = 1$) | $> 10^4$ | $79.3 \pm 3.7$ | $16.7 \pm 0.1$ | $0.001 \pm 0.000$ | $15.0 \pm 0.6$ |
| **ENFORCE** | $-27.77 \pm 0.02$ | $0.0 \pm 0.0$ | $0.0 \pm 0.0$ | $0.14 \pm 0.08$ | $69.4 \pm 23.1$ |

split remain unchanged. We consider a problem with 200 variables and 150 nonlinear constraints. The results are reported in Table 2. ENFORCE successfully predicts optimal solutions satisfying the set of nonlinear constraints across the whole test set. ENFORCE consistently achieves a 25× speedup in inference compared to the nonlinear programming solver, while maintaining the optimal objective within $6\%$. Traditional deep learning and soft constraint methods perform poorly when faced with nonlinear constraints, resulting in significant infeasibility or failure to approximate an optimal solution. Note that a comparison with DC3 is not included, as the original implementation does not support large-scale problem benchmarking involving nonlinear constraints. Once again, a scalability analysis across multiple problem dimensions is reported in Appendix D (Table 6 and 7).

## 5.2 REAL-WORLD CASE STUDY

We evaluate ENFORCE on the real-world engineering case study introduced by Iftakher et al. (2025) and described in Appendix D.4. They propose KKT-Hardnet, a neural network surrogate designed to approximate simulation data of a chemical process while enforcing nonlinear physical constraints. To compare to this baseline, we use the same dataset published by the authors (cf. Iftakher et al. (2025) for data generation details), the same backbone architecture (1 hidden layer with 64 neurons), and training parameters (1200 epochs, learning rate of $10^{-3}$). In Table 3, we compare ENFORCE against the results published by the authors. When enforcing nonlinear constraints on the simulation of chemical processes, ENFORCE results in several orders of magnitude more accurate predictions and a speed up of about 100x during inference and training, compared to the recent model KKT-Hardnet.

Table 3: Real-world case study on a chemical process simulation with nonlinear physical constraints. ENFORCE outperforms the baseline method KKT-Hardnet in terms of accuracy and computational time at training and inference, while strictly enforcing the nonlinear physical constraints.

| Method | MSE | Max eq. $\cdot 10^{-2}$ | Mean eq. $\cdot 10^{-3}$ | Inference [s] | Training [min] |
|---|---|---|---|---|---|
| MLP | $(1.5 \pm 1.4) \cdot 10^{-5}$ | $3.8 \pm 1.6$ | $6.1 \pm 1.8$ | $(1.0 \pm 0.0) \cdot 10^{-4}$ | $2.0 \pm 0.0$ |
| KKT-Hardnet | $1.1 \cdot 10^{-1}$ | $0.00$ | $0.00$ | $0.998$ | $194.5$ |
| **ENFORCE** | $(8.0 \pm 7.9) \cdot 10^{-6}$ | $0.00 \pm 0.00$ | $0.00 \pm 0.00$ | $0.01 \pm 0.00$ | $7.5 \pm 0.5$ |

## 6 CONCLUSIONS

We propose ENFORCE, a method to ensure that neural network predictions satisfy a set of $\mathcal{C}^1$ nonlinear constraints $c(x, y) = 0$, without relying on external solvers or incurring significant computational overhead. We prove the stability of the proposed projection mapping during training and provide theoretical insights into its convergence properties and applicability. The effectiveness of the method is demonstrated on (i) large-scale nonconvex optimization problems with nonlinear constraints and (ii) a real-world engineering simulation. Our findings show that (1) ENFORCE consistently achieves constraint feasibility up to specified tolerance in the case studies, (2) task performance improves (up to 40%) when using ENFORCE over baseline methods, and (3) ENFORCE accelerates ($\times 25$) computationally intensive tasks such as constrained optimization.

## REPRODUCIBILITY STATEMENT

All experimental results are fully reproducible. The datasets, source code, model architecture, training strategy, and analysis are provided in the supplemental material, along with model checkpoints and results. The anonymized repository includes clear instructions and configuration files to replicate experiments, while Sections 5 and Appendix D of the paper describe the data generation process, data splitting, optimizer, and training hyperparameters in detail.

## LLM USAGE DECLARATION

The use of LLMs in this paper was limited to writing, editing, and formatting tasks (e.g., grammar and spelling improvements), while they did not impact the proposed core methodology.

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

# A  APPENDIX

# B  MATHEMATICAL DERIVATIONS

This section provides the key mathematical derivations, theorems, and proofs underlying the proposed method. These derivations are intended to illustrate the theoretical foundations of the approach and to support some of the discussions presented in the main text.

## B.1  CLOSED-FORM NEURAL PROJECTION

We derive here the closed-form expression defining a neural projection layer in Def. 1 (Section 4.1). Given the linearized projection problem in Eq. 3, we can define the Lagrangian function as:

$$\mathcal{L}(x, y, \lambda) = \frac{1}{2}(y - \hat{y})^T(y - \hat{y}) + \lambda^T \left( c(x, \hat{y}) + J_y c|_{x,\hat{y}} (y - \hat{y}) \right) \quad (7)$$

Then, a local optimum can be found by solving the KKT conditions (i.e., primal and dual feasibility):

$$\begin{aligned} \nabla_y \mathcal{L} &= (y - \hat{y}) + J_y^T c\big|_{x,\hat{y}} \lambda = 0 \\ c(x, \hat{y}) &+ J_y c|_{x,\hat{y}} (y - \hat{y}) = 0 \end{aligned} \quad (8)$$

To simplify the notation, we define the linear system as:

$$
\begin{aligned}
(1) \quad & (y - \hat{y}) + B^T \lambda = 0 \\
(2) \quad & By - v = 0
\end{aligned}
\tag{9}
$$

where:

$$
\begin{aligned}
B &= \left. J_y c \right|_{x,\hat{y}} \in \mathbb{R}^{N_C \times N_O} \\
v &= \left. J_y c \right|_{x,\hat{y}} \hat{y} - c(x,\hat{y}) \in \mathbb{R}^{N_C}
\end{aligned}
\tag{10}
$$

Solving the system, we obtain a closed form for the neural projection layer:

$$
\tilde{y} = (I - B^T (BB^T)^{-1} B)\hat{y} + B^T (BB^T)^{-1} v
\tag{11}
$$

## B.2   DEVIATION FROM NEWTON'S METHOD

To support the discussion raised in Section 4.1, we show how our method deviates from standard Newton's method for solving nonlinear KKT conditions. Given a nonlinear program:

$$
\begin{aligned}
\tilde{y} = \arg\min_{y} \quad & \frac{1}{2} \|y - \hat{y}\|^2 \\
\text{s.t.} \quad & c(x,y) = 0
\end{aligned}
\tag{12}
$$

With associated Lagrangian function:

$$
\mathcal{L}(x,y,\lambda) = \frac{1}{2}(y - \hat{y})^T (y - \hat{y}) + \lambda^T c(x,y)
\tag{13}
$$

The primal and dual feasibility can be derived as:

$$
\begin{aligned}
\nabla_y \mathcal{L} &= (y - \hat{y}) + J_y^T c(x,y)\lambda = 0 \\
c(x,y) &= 0
\end{aligned}
\tag{14}
$$

Linearizing the system according to Newton's iteration at $y = y_0$ results in:

$$
\begin{aligned}
& (y_0 - \hat{y}) + \left. J_y^T c \right|_{y_0,\lambda_0} \lambda_0 + (y - y_0) + \lambda^T \left. \mathbf{H}_y c \right|_{y_0,\lambda_0} (y - y_0) + \left. J_y^T c \right|_{y_0,\lambda_0} (\lambda - \lambda_0) = 0 \\
& c(x,y_0) + \left. J_y c \right|_{y_0,\lambda_0} (y - y_0) = 0
\end{aligned}
\tag{15}
$$

Thus, assuming to center the linearization in the neural network prediction, i.e., $y_0 = \hat{y}$, and choosing $\lambda_0 = 0$:

$$
\begin{aligned}
& (y - \hat{y}) + \lambda^T \left. \mathbf{H}_y c \right|_{\hat{y},0} (y - \hat{y}) + \left. J_y^T c \right|_{\hat{y},0} \lambda = 0 \\
& c(x,\hat{y}) + \left. J_y c \right|_{\hat{y},0} (y - \hat{y}) = 0
\end{aligned}
\tag{16}
$$

We can conclude that, essentially, our NP layer solves a similar linear system (Eq. 8) which does not comprise the term $\lambda^T \left. \mathbf{H}_y c \right|_{\hat{y},0} (y - \hat{y})$, hence avoiding the computation of the Hessian tensor $\mathbf{H}_y c$. Here, we note some similarity with Gauss-Newton methods used to solve least square problems (Nocedal & Wright, 2006).

## B.3   LOCAL CONVERGENCE RATE

The projection operator $\mathcal{P}$ solves an SQP subproblem of the form:

$$
\begin{aligned}
\min_{y} \quad & \frac{1}{2}(y - \hat{y})^T \bar{H}(y - \hat{y}) \\
\text{s.t.} \quad & J_y c (y - \hat{y}) + c(x,\hat{y}) = 0,
\end{aligned}
\tag{17}
$$

which approximates the original nonlinear program:

$$\min_y \frac{1}{2}(y - \hat{y})^T I(y - \hat{y})$$
$$\text{s.t.} \quad c(x, y) = 0 \tag{18}$$

The Hessian $\bar{H}$, as observed in the deviation from Newton's method B.2, does not include the second-order derivatives of the constraints that would appear in the full Lagrangian Hessian. The resulting method is often called *Gauss-Newton SQP step*, since the way the constraints derivatives are dropped reminds of the Gauss-Newton method for nonlinear least squares (Nocedal & Wright, 2006).

Supported by SQP theory (Nocedal & Wright, 2006), conditions for local convergence can be derived. We assume $y^*$ to be a local solution to the original nonlinear program (Eq.1) at which the following conditions hold (Nocedal & Wright, 2006):

H1 The objective function and the constraints are twice differentiable in a neighborhood of $y^*$ with Lipschitz continuous second derivatives.

H2 The linear independence constraint qualification (LICQ) holds at $y^*$. Then, the KKT conditions are satisfied for a vector of Lagrangian multipliers $\lambda^*$.

H3 The second-order sufficient conditions (SOSC) hold at $(y^*, \lambda^*)$.

The KKT conditions for the original nonlinear program are defined as:

$$F(z) = \begin{bmatrix} \nabla_y \mathcal{L}(y, \lambda) \\ c(y) \end{bmatrix}, \quad \text{with} \quad z = \begin{bmatrix} y \\ \lambda \end{bmatrix}, \tag{19}$$

and are satisfied by a vector $z^* = [y^* \quad \lambda^*]^T$.

We define the Jacobian of the KKT conditions of the original nonlinear program (Eq. 18) in a neighborhood of the local solution as:

$$J^{(k)} = \begin{bmatrix} \nabla_{yy}^2 \mathcal{L}^{(k)} & J_c^T \\ J_c & 0 \end{bmatrix} \tag{20}$$

We assume that LICQ and SOQC hold also in the neighborhood of $z^*$ (H2 and H3), hence the Jacobian at iteration $k$, $J^{(k)}$, is non-singular and thus invertible.

The deviation of the projection operator $\mathcal{P}$ from the complete SQP step can be expressed through a matrix $E$ holding the second-order derivatives of the constraints:

$$E = \begin{bmatrix} \sum_i \lambda_i^{(k)} \nabla^2 c_i(y^{(k)}) & 0 \\ 0 & 0 \end{bmatrix}, \tag{21}$$

such that $J^{(k)} = \bar{J} + E$, with $\bar{J}$ being the Jacobian of the KKT conditions associated with the problem in Eq 17.

At iteration $k$, we define the residual $r^{(k)} = F(z^{(k)})$, the error $e^{(k)} = z^{(k)} - z^*$ and solve for the Newton's step $s^{(k)}$:

$$\bar{J}s^{(k)} = -r^{(k)} \qquad \text{(QP solve, Newton step)}$$
$$z^{(k+1)} = z^{(k)} + s^{(k)} \tag{22}$$
$$e^{(k+1)} = e^{(k)} + s^{(k)}$$

Since $F(z)$ is twice continuously differentiable, using Taylor expansion:

$$F(z^{(k)}) = J^{(k)}e^{(k)} + r^{(k)}, \qquad \text{with} \quad r^{(k)} = \mathcal{O}(||e^{(k)}||^2) \tag{23}$$

Thus, a QP step can be expressed as:

$$\bar{J}s^{(k)} = -r^{(k)} = -F(z^{(k)}) = -J^{(k)}e^{(k)} - r^{(k)} \tag{24}$$

From the definition of the Jacobian $\bar{J}$ and given that $J^{(k)}$ is invertible:

$$(J^{(k)} - E)s^{(k)} = -J^{(k)}e^{(k)} - r^{(k)} \tag{25}$$

$$((J^{(k)})^{-1}J^{(k)} - (J^{(k)})^{-1}E)s^{(k)} = -(J^{(k)})^{-1}J^{(k)}e^{(k)} - (J^{(k)})^{-1}r^{(k)} \tag{26}$$

$$(I - M)s^{(k)} = -e^{(k)} - (J^{(k)})^{-1}r^{(k)}, \tag{27}$$

$$s^{(k)} = -(I - M)^{-1}(e^{(k)} + (J^{(k)})^{-1}r^{(k)}), \tag{28}$$

with $M = (J^{(k)})^{-1}E$.

Thus, in the neighborhood of the solution, the error at iteration $k + 1$ can be expressed as:

$$e^{(k+1)} = z^{(k+1)} - z^* = e^{(k)} + s^{(k)} = e^{(k)} - (I - M)^{-1}(e^{(k)} + (J^{(k)})^{-1}r^{(k)}) \tag{29}$$

Rearranging:

$$\begin{aligned}
e^{(k+1)} &= (I - (I - M)^{-1})e^{(k)} - (I - M)^{-1}(J^{(k)})^{-1}r^{(k)} \\
&= (I - M)^{-1}((I - M) - I)e^{(k)} - (I - M)^{-1}(J^{(k)})^{-1}r^{(k)} \\
&= -(I - M)^{-1}Me^{(k)} - (I - M)^{-1}(J^{(k)})^{-1}r^{(k)}
\end{aligned} \tag{30}$$

Banach's lemma then gives:

$$||(I - M)^{-1}|| \leq \frac{1}{1 - ||M||} = \frac{1}{1 - \rho} = C_0 \tag{31}$$

Then we can estimate the error:

$$||e^{(k+1)}|| \leq C_0(||M|| \, ||e^{(k)}|| + ||(J^{(k)})^{-1}|| \, ||r^{(k)}||) \tag{32}$$

Since $r^{(k)} = O(||e^{(k)}||^2), \exists \, C_1 > 0 : ||r^{(k)}|| \leq C_1||e^{(k)}||^2$, then:

$$||e^{(k+1)}|| \leq C_0||M|| \, ||e^{(k)}|| + C_0||(J^{(k)})^{-1}|| \, C_1||e^{(k)}||^2) \tag{33}$$

We can conclude that, in the neighborhood of the solution:

- If $M = 0$, the linear term vanishes and yields quadratic convergence, i.e., when the constraints are affine and thus the second order derivative of the constraints vanish ($\nabla^2 c_i = 0$).
- If $M \neq 0$ but $||M|| < 1$, it is guaranteed strictly linear convergence with rate $||M||$, plus a higher-order correction.
- If $||M|| \geq 1$, the Gauss–Newton step alone may not converge. Second-order corrections (or using the full Lagrangian Hessian) are then required.

Among state-of-the-art methods for constrained learning, we recognize that Newton-based completion approaches exhibit a quadratic convergence rate. However, as will be shown below (Appendix B.6), they can suffer from training instabilities.

### B.4 UNIQUENESS OF THE PROJECTION

Given a set of constraints $c(y) = 0$, sufficiently smooth and with full rank matrix $J_y c$ when $c(y) = 0$, we define the $(N_O - N_C)$-dimensional submanifold $\mathcal{M} = \{y \in \mathbb{R}^{N_O} : c(y) = 0\}$ in the ambient-space manifold $\mathcal{N} \in \mathbb{R}^{N_O}$. Then, we can prove that in the neighborhood of the manifold, the orthogonal projection is unique despite the nonlinearity of the constraints.

**Definition 3** (Tubular neighbourhood). *Let $\mathcal{M}$ be a smooth embedded submanifold of a smooth manifold $\mathcal{N}$ and let $\nu(\mathcal{M}) \to \mathcal{M}$ be its normal bundle. A* tubular neighbourhood *of $\mathcal{M}$ in $\mathcal{N}$ is an open set $U \subseteq \mathcal{N}$ for which there exists an open neighbourhood $\mathcal{V}$ of the zero section in $\nu(\mathcal{M})$ and a diffeomorphism $\Phi$:*

$$\Phi : \mathcal{V} \longrightarrow U$$

*that restricts to the identity on the zero section. In other words, $U$ is obtained by smoothly "thickening" $\mathcal{M}$ along its normal directions.*

**Definition 4** (Reach of a manifold (Federer, 1959, Def. 4.1)). *The reach of a closed subset $\mathcal{M}$ (in particular, an embedded submanifold),* $\text{reach}(\mathcal{M})$, *is the largest radius $\rho > 0$ for which $\forall \hat{y}$ such that $\text{dist}(\hat{y}, \mathcal{M})$ there is a unique nearest point $\tilde{y} \in \mathcal{M}$. Formally:*

$$U_\rho := \left\{ y \in \mathbb{R}^{N_O} : \text{dist}(y, \mathcal{M}) < \rho \right\}.$$

*For every $\rho > 0$ define the nearest–point (metric) projection:*

$$\mathcal{P}_\rho : U_\rho \longrightarrow \mathcal{M}, \qquad \mathcal{P}_\rho(y) = \arg\min_{y \in \mathcal{M}} \|y - \hat{y}\|.$$

*Since $\mathcal{M}$ is closed, the minimum exists, but it may fail to be unique.*

*The* reach *of $\mathcal{M}$ is the supremum $\rho$ such that there is a single minimum to the projection above:*

$$\text{reach}(\mathcal{M}) := \sup \left\{ \rho > 0 : \mathcal{P}_\rho \text{ is well-defined (single-valued)} \right\}.$$

For instance, $\text{reach}(\mathcal{M}) = \infty$ exactly when the projection $\mathcal{P}_\rho$ is single-valued for every $\rho > 0$, e.g. when $\mathcal{M}$ is an affine subspace.

Hence, we can derive conditions for the uniqueness of the projection.

**Theorem 1** (Uniqueness of the projection). *Given a prediction $\hat{y} \in \mathbb{R}^{N_O}$ and a smooth constraints $c(y) : \mathbb{R}^{N_O} \to \mathbb{R}^N_C$ with full rank Jacobian $J_y c$ when the constraints are satisfied, defining a submanifold $\mathcal{M} = \{y \in \mathbb{R}^{N_O} : c(y) = 0\}$, if $\text{dist}(\hat{y}, \mathcal{M}) < \text{reach}(\mathcal{M})$, then the minimizer $\tilde{y}$ defined as:*

$$\tilde{y} = \arg\min_{y \in \mathcal{M}} \frac{1}{2} \|y - \hat{y}\|.$$

*exists and is unique.*

**Proof:** The proof follows easily from Definition 3 and Definition 4.

The reach depends on the geometry of the constraint set (e.g., the reach is infinite for linear constraints and approaches zero near sharp corners or singularities). Therefore, for the projection to be well-defined, $\hat{y}$ must lie within a sufficiently small tubular neighborhood of $\mathcal{M}$.

### B.5 No-Worse prediction

We prove that when a unique projection exists on a convex constraint manifold with positive reach, if $y^*$ is the ground-truth, then the projected prediction $\tilde{y}$ is *always* a better prediction than the original prediction $\hat{y}$.

**Proposition 2** (No-worse property). *Let $\mathcal{M} \subset \mathbb{R}^{N_O}$ be a smooth, convex, embedded submanifold defined by equality constraints $c(y) = 0$, with $c : \mathbb{R}^{N_O} \to \mathbb{R}^{N_C}$ smooth and $J_y c$ full rank and Lipschitzian on $\mathcal{M}$ (hence, $\mathcal{M}$ has positive reach (Federer, 1959)). Let $\text{reach}(\mathcal{M}) > 0$.*

*Suppose:*

- *$y^* \in \mathcal{M}$ is the (unknown) ground truth,*

- *$\hat{y} \in \mathbb{R}^{N_O}$ is a model prediction such that $\text{dist}(\hat{y}, \mathcal{M}) < \text{reach}(\mathcal{M})$,*

- *$\tilde{y} := \mathcal{P}(\hat{y})$ is the unique orthogonal projection of $\hat{y}$ onto $\mathcal{M}$.*

*Then:*

$$\|\tilde{y} - y^*\| \leq \|\hat{y} - y^*\|,$$

*with equality if and only if $\hat{y} \in \mathcal{M}$. That is, projecting onto the constraint manifold never increases the Euclidean error with respect to the ground truth, and strictly reduces it when the constraints are violated.*

**Proof:** Since $\hat{y} \in U_\rho := \{y \in \mathbb{R}^{N_O} : \mathrm{dist}(y, \mathcal{M}) < \mathrm{reach}(\mathcal{M})\}$, the orthogonal projection $\mathcal{P}(\hat{y})$ is well-defined and unique. Denote $\tilde{y} := \mathcal{P}(\hat{y})$ and recall that $\tilde{y}$ is the closest point to $\hat{y}$ on $\mathcal{M}$.

By the normality property of projections onto manifolds with positive reach (cf. (Federer, 1959, Def. 4.1)), we have:

$$\langle \hat{y} - \tilde{y}, y^* - \tilde{y} \rangle \geq 0 \quad \text{for all } y^* \in \mathcal{M}. \tag{34}$$

Expanding the squared distance yields:

$$\|\hat{y} - y^*\|^2 = \|\tilde{y} - y^* + \hat{y} - \tilde{y}\|^2$$
$$= \|\tilde{y} - y^*\|^2 + \|\hat{y} - \tilde{y}\|^2 + 2\langle \hat{y} - \tilde{y}, \tilde{y} - y^* \rangle.$$

Using the projection property $\langle \hat{y} - \tilde{y}, y^* - \tilde{y} \rangle \geq 0$, we get:

$$\|\hat{y} - y^*\|^2 \geq \|\tilde{y} - y^*\|^2 + \|\hat{y} - \tilde{y}\|^2, \tag{35}$$

and hence:

$$\|\tilde{y} - y^*\|^2 \leq \|\tilde{y} - y^*\|^2 - \|\hat{y} - \tilde{y}\|^2$$
$$\|\tilde{y} - y^*\|^2 \leq \|\tilde{y} - y^*\|^2 \tag{36}$$
$$\|\tilde{y} - y^*\| \leq \|\hat{y} - y^*\|$$

Equality holds if and only if $\|\hat{y} - \tilde{y}\| = 0$, i.e., $\hat{y} = \tilde{y} \in \mathcal{M}$.

### B.6    CONDITIONING OF THE PROJECTION OPERATION

The training stability and robustness can potentially be influenced by layers appended on top of a neural network (e.g., projection operation or null-space completion). We demonstrate that, in the neighborhood of the constraints manifold, and under mild assumptions, the projection operation is a *non-expansive* mapping in the Banach sense, i.e., its Lipschitz constant $L \leq 1$. This implies stability and adversarial robustness during the forward pass and well-conditioned gradient-flow dynamics during the backward pass. On the other hand, null-space completion methods (i.e., in DC3 and other methods (Donti et al., 2021; Beucler et al., 2019)) are characterized by a Lipschitz constant dependent on the Jacobian of the constraints. This can lead to training instabilities, vanishing or exploding gradients, whenever the Jacobian is ill-conditioned.

**Theorem 2** (Non-expansiveness of the projection operator)**.** *Given the nonlinear program:*

$$\tilde{y} = \arg\min_y \frac{1}{2}\|y - \hat{y}\|^2 \quad s.t. \quad c(y) = 0, \tag{37}$$

*with $\hat{y} \in \mathbb{R}^{N_O}$, $\tilde{y} \in \mathbb{R}^{N_O}$, and $c(y) : \mathbb{R}^{N_O} \to \mathbb{R}^{N_C}$ smooth, continuous constraints with a full-rank Jacobian $J_y c$ and being the set $C = \{y \in \mathbb{R}^{N_O} : c(y) = 0\}$ convex in the vector space of the neural network output $\mathbb{R}^{N_O}$ (i.e., $c(y)$ is affine), the projection operator $\mathcal{P} : \tilde{y} = \mathcal{P}(\hat{y})$ solving the nonlinear program (Eq. 37) is a non-expansive mapping in the neighborhood of the constraints manifold:*

$$\forall \hat{y}_i, \hat{y}_j \in \mathbb{R}^{N_O}, \quad \mathrm{dist}(\hat{y}_i, \mathcal{M}) < \mathrm{reach}(\mathcal{M}) \wedge \mathrm{dist}(\hat{y}_j, \mathcal{M}) < \mathrm{reach}(\mathcal{M})$$
$$\Rightarrow \|\mathcal{P}(\hat{y}_i) - \mathcal{P}(\hat{y}_j)\| \leq \|\hat{y}_i - \hat{y}_j\| \tag{38}$$

**Proof:** The proof begins by formally proving that $\mathcal{P}$ is an orthogonal projection operator, by studying its dynamics with respect to $\hat{y}$.

The Lagrangian of the problem 12 can be expressed as:

$$\mathcal{L}(y, \lambda) = \frac{1}{2}\|y - \hat{y}\|^2 + \lambda^T c(y) \tag{39}$$

At the optimal point $\tilde{y} = \mathcal{P}(\hat{y})$:

$$\begin{cases} y - \hat{y} + J_y c^T \lambda = 0 \\ c(y) = 0 \end{cases} \tag{40}$$

We now consider $y$ and $\lambda$ as implicit functions of $\hat{y}$:

$$y = y(\hat{y}), \quad \lambda = \lambda(\hat{y}) \quad \Rightarrow F(y(\hat{y}), \lambda(\hat{y}), \hat{y}) = \begin{cases} y(\hat{y}) - \hat{y} + J_{y(\hat{y})} c^T \lambda(\hat{y}) = 0 \\ c(y(\hat{y})) = 0 \end{cases} \tag{41}$$

The total derivative of $F$ can be derived as:

$$\frac{d}{d\hat{y}}(F) = \frac{dF}{dy}\Big|_{\hat{y},\lambda} \frac{dy}{d\hat{y}} + \frac{\partial F}{\partial \lambda}\Big|_{\hat{y},y} \frac{d\lambda}{d\hat{y}} + \frac{dF}{d\hat{y}}\Big|_{y,\lambda}, \tag{42}$$

where:

$$\frac{dF}{dy}\Big|_{\hat{y},\lambda} = \begin{bmatrix} I + \frac{d}{dy}\left(J_{y(\hat{y})} c^T \lambda(\hat{y})\right) \\ J_y c \end{bmatrix} = \begin{bmatrix} I + H \\ J_y c \end{bmatrix} \tag{43}$$

$$\frac{\partial F}{\partial \lambda}\Big|_{\hat{y},y} = \begin{bmatrix} J_y c^T \\ 0 \end{bmatrix} \tag{44}$$

$$\frac{dF}{d\hat{y}}\Big|_{y,\lambda} = \begin{bmatrix} -I \\ 0 \end{bmatrix} \tag{45}$$

Rearranging in (block) matrix form:

$$\begin{bmatrix} \frac{dF}{dy}\Big|_{\hat{y},\lambda} & \frac{\partial F}{\partial \lambda}\Big|_{\hat{y},y} \end{bmatrix} \begin{bmatrix} \frac{dy}{d\hat{y}} \\ \frac{d\lambda}{d\hat{y}} \end{bmatrix} = -\begin{bmatrix} \frac{dF}{d\hat{y}}\Big|_{y,\lambda} \end{bmatrix} \tag{46}$$

$$\begin{bmatrix} I + H & J_y c^T \\ J_y c & 0 \end{bmatrix} \begin{bmatrix} \frac{dy}{d\hat{y}} \\ \frac{d\lambda}{d\hat{y}} \end{bmatrix} = \begin{bmatrix} I \\ 0 \end{bmatrix} \tag{47}$$

We simplify the notation to solve the linear system by defining:

$$Z := \frac{dy}{d\hat{y}} \in \mathbb{R}^{N_O \times N_O}$$
$$L := \frac{d\lambda}{d\hat{y}} \in \mathbb{R}^{N_C \times N_O} \tag{48}$$
$$J := J_y c \in \mathbb{R}^{N_C \times N_O}$$

Note that the row space of $J$ is the normal space to the constraints, while its kernel (column space) is the tangent space.
Then:

$$\begin{cases} (I + H)Z + J^T L = I \\ JZ = 0 \end{cases} \tag{49}$$

The second equation means that every column of $Z$ lives in $\ker(J)$, i.e., the tangent space. Then, since $\hat{y}$ is in the reach of the constraint manifold, let $P := I - J^T (JJ^T)^{-1} J$ be the *unique projector* onto the subspace $\ker(J)$. $P$ satisfies the following properties:

- $P^2 = P \Rightarrow$ idempotent
- $P^T = P \Rightarrow$ symmetric
- $JP = 0 \Rightarrow \mathrm{Im}(P) = \ker(J)$ ($\mathrm{Im}(\cdot)$ is the *image* and $\ker(\cdot)$ is the *kernel*)

Then, any solution to the system in Eq. 49 must satisfy $Z = PZ$.

**Sub-proof:** $Jz = 0 \Rightarrow$ each column $z^{(k)}$ lies in $\ker(J)$. Then:

$$PZ = (I - J^T(JJ^T)^{-1}J)Z = Z - J^T(JJ^T)^{-1}JZ \stackrel{JZ=0}{=} Z \tag{50}$$

Now, assuming convexity for the set $C = \{y \in \mathbb{R}^{N_O} : c(y) = 0\}$, we can guess the simplest representative of $Z$ of the form of $PZ$, such as $Z = P$. Then, from the first equation and using the definition of the projector matrix:

$$(I + H)P + J^T L = I = P + J^T(JJ^T)^{-1}J = P + Q, \tag{51}$$

where $Q = J^T(JJ^T)^{-1}J$ is the normal component of the identity matrix. Developing the algebra:

$$IP + HP + J^T L = P + Q \tag{52}$$

$$J^T L = Q - HP \tag{53}$$

Since $J$ has full rank, there exists a left-inverse (Moore-Penrose pseudoinverse) $J^+$ such that $J^+ J = I$ and $J^+ = (JJ^T)^{-1}J$. Hence, solving for $L$ and using the definition of $Q$:

$$\begin{aligned}
L &= J^+(Q - HP) \\
&= (JJ^T)^{-1}JQ - (JJ^T)^{-1}JHP \\
&= (JJ^T)^{-1}JJ^T(JJ^T)^{-1}J - (JJ^T)^{-1}JHP \\
&= (JJ^T)^{-1}J(I - HP)
\end{aligned} \tag{54}$$

We can conclude:

$$\begin{aligned}
Z &= \frac{dy}{d\hat{y}} = I - J^T(JJ^T)^{-1}J \\
L &= \frac{d\lambda}{d\hat{y}} = (JJ^T)^{-1}J(I - HP)
\end{aligned} \tag{55}$$

Considering now the projection operator $\mathcal{P}$ such that $y = \mathcal{P}(\hat{y})$, we found that the Jacobian of such operator is the orthogonal projector:

$$\frac{dy}{d\hat{y}} = J_{\hat{y}}\mathcal{P} = I - J^T(JJ^T)^{-1}J \tag{56}$$

Hence, $J_{\hat{y}}\mathcal{P}$ is idempotent, symmetric, and does not depend on the second-order derivatives of the constraints in $H$. We can easily prove that all the eigenvalues of a symmetric and idempotent matrix are either 0 or 1. Thus, the spectral norm of the Jacobian is 1:

$$||J_{\hat{y}}\mathcal{P}|| = \max\{|\lambda_p|\} = 1, \tag{57}$$

with $\lambda_i$ being the $p$-th eigenvalue (to be not confused with the Lagrangian multipliers). Being the Jacobian bounded to 1, the projection operator $\mathcal{P}$ is a *non-expansive* mapping in $\mathbb{R}^{N_O}$. In other words, the mapping operated by $\mathcal{P}$ is 1-Lipschitz:

$$||\mathcal{P}(\hat{y}_i) - \mathcal{P}(\hat{y}_j)|| \leq ||\hat{y}_i - \hat{y}_j||, \tag{58}$$

which concludes the proof of the theorem.

**Practical implications** The bounding of the projection operator leads to important practical consequences in training a neural network, such as:

- *Adversarial robustness* (or forward sensitivity): Small input perturbation cannot be magnified by the projection operation, which follows directly from Eq. 58.

- *Stability of gradient flow dynamics* (or backward sensitivity): During backpropagation, the pre-projection gradient is multiplied by a factor $J_{\hat{y}}\mathcal{P}^T = J_{\hat{y}}\mathcal{P}$, which is bounded.

$$\nabla_{\hat{y}}\ell = J_{\hat{y}}\mathcal{P}^T \nabla_{\tilde{y}}\ell \quad \text{with} \quad ||\nabla_{\hat{y}}\ell|| \le ||J_{\hat{y}}\mathcal{P}^T||\, ||\nabla_{\tilde{y}}\ell||, \tag{59}$$

with $\ell$ being the loss term. Hence, the gradient flow is well conditioned (no exploding gradients can originate from the projection step, the scale of pre-projection and post-projection gradients is comparable), which leads to smoother and faster training convergence.

**Comparison with null-space methods (predict-and-complete)** In null-space methods, a neural network $f_\theta$ outputs a vector $z \in \mathbb{R}^{N_O - N_C} : z = f_\theta(x)$. Then, a mapping $\varphi_x$ uses the constraints $c(x, y)$ to *complete* the output vector with $\varphi_x(z) \in \mathbb{R}^{N_C}$ such that $y = \begin{bmatrix} z & \varphi_x(z) \end{bmatrix}^T$.
The Jacobian of the constraints can be factored into two blocks:

$$Jc = \begin{bmatrix} Jc_{[0:m]} & Jc_{[m:N_O]} \end{bmatrix}, \quad \text{with} \quad m = N_O - N_C \tag{60}$$

The mapping $\varphi_x$ represents the solution of a (non)linear system of equations, either explicitly or by using a root-finding solver (e.g., Newton method). According to Donti et al. (2021), the differential of such mapping can be computed leveraging the implicit function theorem (as in OptNet (Amos & Kolter, 2017)):

$$\frac{\partial \varphi_x}{\partial z} = J_z\varphi = -(Jc_{[m:N_O]})^{-1}Jc_{[0:m]} \tag{61}$$

Hence, the operator-norm bound leads to:

$$||J_z\varphi|| \le ||(Jc_{[m:N_O]})^{-1}||\, ||Jc_{[0:m]}|| \tag{62}$$

$$\|\varphi_{x1} - \varphi_{x2}\| \le ||J_z\varphi||\, \|z_1 - z_2\|, \tag{63}$$

We observe that the Jacobian of the mapping (completion step) is not bounded, as it depends directly on the constraints. Indeed, while the second term $||Jc_{[0:m]}||$ can be tuned by re-scaling the variables in the neural network, there is no simple trick to condition the term $||(Jc_{[m:N_O]})^{-1}||$. The spectral norm of the inverse matrix can be approximated by the smallest singular value $\sigma_{min}(Jc_{[m:N_O]})$. Then, if the constraints Jacobian matrix block $Jc_{[m:N_O]}$ is not well-conditioned:

$$||(Jc_{[m:N_O]})^{-1}|| \sim \frac{1}{\sigma_{min}}$$
$$\sigma_{min} \to 0, \quad ||J_z\varphi|| \to \infty \tag{64}$$

Thus, we conclude that the completion step is in general *not* 1-Lipschitz, with Lipschitz constant $\kappa = ||(Jc_{[m:N_O]})^{-1}||$. This can potentially lead to instabilities during training and result in sub-accurate NNs, with respect to unconstrained counterparts (e.g., as in (Beucler et al., 2019)):

- *Adversarial robustness* (or forward sensitivity): Small input perturbation can be amplified by a factor $\kappa$.

- *Stability of gradient flow dynamics* (or backward sensitivity): During backpropagation, according to (Donti et al., 2021), the upstream gradient can be computed as:

$$\frac{d\ell}{dz} = \frac{\partial \ell}{\partial z} - \frac{\partial \ell}{\partial \varphi_x} \cdot \frac{\partial \varphi_x}{\partial z} \tag{65}$$

We can bound the upstream loss as:

$$\left|\left\|\frac{\partial\ell}{\partial z}\right\| - \left\|\frac{\partial\ell}{\partial\varphi_x}\right\|\left\|\frac{\partial\varphi_x}{\partial z}\right\|\right| \le \left\|\frac{d\ell}{dz}\right\| \le \left\|\frac{\partial\ell}{\partial z}\right\| + \left\|\frac{\partial\ell}{\partial\varphi_x}\right\|\left\|\frac{\partial\varphi_x}{\partial z}\right\| \tag{66}$$

And considering the derived Lipschitz constant $\kappa$:

$$\left|\left\|\frac{\partial\ell}{\partial z}\right\| - \kappa\left\|\frac{\partial\ell}{\partial\varphi_x}\right\|\right| \le \left\|\frac{d\ell}{dz}\right\| \le \left\|\frac{\partial\ell}{\partial z}\right\| + \kappa\left\|\frac{\partial\ell}{\partial\varphi_x}\right\| \tag{67}$$

Hence, the gradients expand or shrink by a factor $\kappa$, potentially leading to undesired phenomena such as vanishing or exploding gradients.

## C   IMPLEMENTATION DETAILS

### C.1   ADANP ALGORITHM

Algorithm 1 provides a high-level overview of the procedure underlying the AdaNP module. The depth of AdaNP (i.e., the number of projection iterations) adapts to satisfy the constraints requirements according to a specified tolerance.

---

**Algorithm 1** AdaNP: Adaptive-depth Neural Projection

1: **Input:** input $x$, preliminary prediction $\hat{y}$, constraints $c$, tolerance $\varepsilon_t$, maximum depth $d_{\max}$
2: **Initialize:** depth counter $i \leftarrow 0$
3: **while** $m(c(x,\hat{y})) > \varepsilon_t$ **and** $i < d_{\max}$ **do**
4:     **Compute constraints Jacobian:** $J_y c$
5:     **Compute:** $B = J_y c|_{x,\hat{y}}$
6:     **Compute:** $v = B\hat{y} - c(x,\hat{y})$
7:     **Compute:** $B^* = I - B^T(BB^T)^{-1}B$
8:     **Compute:** $v^* = B^T(BB^T)^{-1}v$
9:     **Project:** $\tilde{y} = B^*\hat{y} + v^*$
10:     **Update:** $\hat{y} = \tilde{y}$
11:     **Increment:** $i \leftarrow i + 1$
12: **end while**
13: **return** $\tilde{y}$

---

Here, $m(c(x,\hat{y}))$ represents some measure of the constraint residual, where $m(\cdot)$ can be the *max* or the *mean* operator.

### C.2   ADANP ACTIVATION ALGORITHM

In Section 4.4, we introduced a strategy for constrained learning during the early stages of training. The algorithm we propose (Algorithm 2) assesses the effectiveness of the projection operation by quantifying and comparing a task-specific loss measure ($m_\ell$) computed on both the preliminary and the projected predictions. This loss measure is different from the complete training loss function. In the presented experiments, denoting by $\bar{y}$ either $\hat{y}$ or $\tilde{y}$, depending on the context, we define:

- **Function fitting:** For regression tasks, $m_\ell = \frac{1}{N}\sum_{i=1}^{N}||y_i - \bar{y}_i||^2$ is the standard mean squared error loss.

- **Constrained optimization problem**: In the context of unsupervised learning for parametric optimization problems, we define the loss measure as $m_\ell = \frac{1}{N}\sum_{i=1}^{N} f_i(x,\bar{y}) + \frac{\lambda_C}{N}\sum_{i=1}^{N}||c(x_i,\bar{y})||$. It is essential to include a penalty term for constraint violations, as strictly enforcing the constraints may inherently lead to larger values of the objective function.

The algorithm is applied at each forward pass during training. Specifically, it is executed after the raw backbone output (preliminary prediction $\hat{y}$) and the adaptive projection step (projected prediction $\tilde{y}$).

To provide an intuitive analogy, this mechanism is reminiscent of trust-region methods in constrained optimization (Nocedal & Wright, 2006), where a candidate step is only *accepted* if it leads to a sufficient improvement in the objective. In our case, a task-specific cost function is used to assess whether the projection improves the prediction.

---

**Algorithm 2** AdaNP activation algorithm during training

---

1: **Input:** neural network $f_\theta$, input $x$, loss measure $m_\ell$
2: **Predict:** $\hat{y} = f_\theta(x)$
3: **Project:** $\tilde{y} = \mathcal{P}(\hat{y})$
4: **if** $m_\ell(\tilde{y}) > m_\ell(\hat{y})$ **then**
5:     $\tilde{y} = \hat{y}$                 ▷ Discard the projection and use original prediction
6: **else**
7:     $\tilde{y}_n = \text{AdaNP}(\tilde{y})$                        ▷ Activate AdaNP
8: **end if**
9: **return** $\tilde{y}_n$

---

### C.3 BATCH LOCAL PROJECTION

The computationally most expensive operation in the neural projection layer is the matrix inversion $(BB^T)^{-1}$ (Def. 1 in Section 4.1), which has a complexity of $\mathcal{O}(N^3)$. At inference time, $N = N_C$, since $B \in \mathbb{R}^{N_C \times N_O}$ and $BB^T \in \mathbb{R}^{N_C \times N_C}$. Hence, considering a number of constraints $N_C < 10^3$, the matrix inversion is performed in less than 1 million FLOPs, which is an affordable amount for most modern CPUs and GPUs. During training, assuming the use of batch gradient descent and defining the batch size $(BS)$ as the number of data points processed in a single iteration, an equivalent number of matrix inversions must be performed. Thus, the computational cost for a single batch is apparently $\mathcal{O}(BS \times N_C^3)$. To address this, we leverage parallel computing on GPUs by constructing a *rank-3* tensor $\mathbf{B} \in \mathbb{R}^{BS \times N_C \times N_O}$ to hold $BS$ local matrices $B$. Similarly, a *rank-2* tensor $V \in \mathbb{R}^{BS \times N_C}$ is built to store $BS$ local vectors $\mathbf{v}$. Modern deep learning libraries enable batch operations, such as matrix inversion, which reduce the effective complexity to $\mathcal{O}(N_C^3)$ (i.e., scaling only with the number of constraints). To invert the batch of matrices, we use the Cholesky factorization algorithm (Burden & Faires, 2005).

Given these conditions and the capabilities of current hardware, the neural projection operation remains computationally efficient even during training when the number of constraints is in the order of a few hundred. Moreover, the complexity of this method is *equivalent* to other state-of-the-art methods such as DC3 (Donti et al., 2021), where each Newton's step in the completion algorithm requires the inversion of a batch of $(N_C \times N_C)$ matrices.

### C.4 MEMORY FOOTPRINT

For each neural projection layer, the AdaNP module creates four 3D tensors:

- Tensor $\mathbf{B}$ of shape $(BS, N_C, N_O)$
- Tensor $\mathbf{b}$ of shape $(BS, N_C, 1)$
- Tensor $\mathbf{B}^*$ of shape $(BS, N_O, N_O)$
- Tensor $\mathbf{b}^*$ of shape $(BS, N_O, 1)$

Assuming 32-bit floating-point representation (i.e., 4 bytes per element), the memory required to store these tensors for a single projection step is:

$$M_{\mathcal{P}} = 4 \cdot BS \cdot (N_O + 1) \cdot (N_C + N_O) \quad \text{(bytes)} \tag{68}$$

Figures 2 and 3 illustrate how the memory requirement for a single projection layer is affected by variations in batch size, the number of predicted variables, and the number of constraints.

The total memory usage of the *unrolled* AdaNP depends on the mode of operation:

- During training, tensors are retained at each of the $n$ projection steps (i.e., for gradient computation), resulting in:

$$M_{\text{train}} = 4 \cdot n \cdot BS \cdot (N_O + 1) \cdot (N_C + N_O) \tag{69}$$

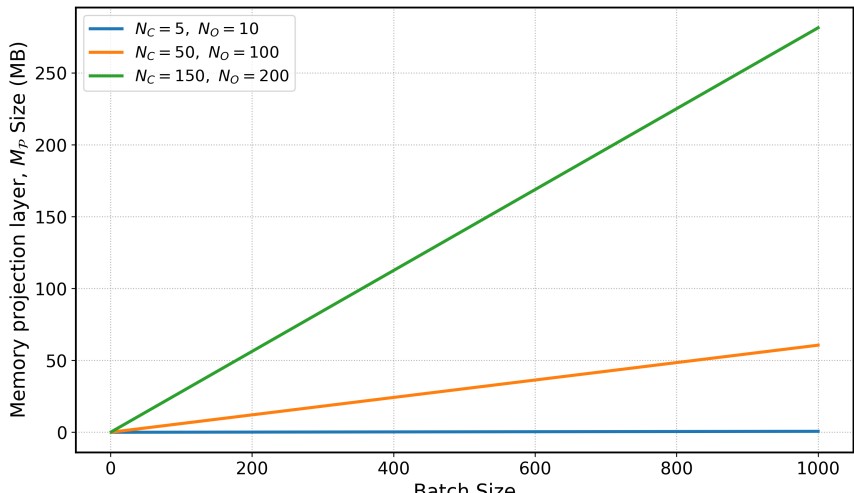

Figure 2: Memory footprint of a single projection layer $\mathcal{P}$. $N_C$ is the number of constraints, and $N_O$ is the dimension of the neural network output. The memory usage scales linearly with the batch size, with the growth rate determined by the number of predicted variables and constraints. For a batch size of 200, the memory requirement is on the order of tens of megabytes.

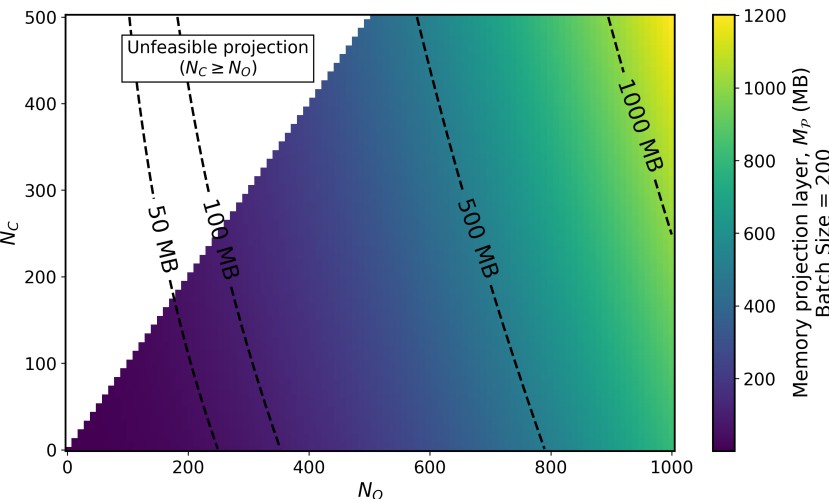

Figure 3: Memory footprint for a fixed batch size of 200 samples. $N_C$ is the number of constraints, and $N_O$ is the dimension of the neural network output. For large-scale tasks involving thousands of variables and constraints, the AdaNP module can become memory-intensive, with each projection layer requiring more than 1 GB of memory.

- During inference, tensors are not retained across steps, and the peak memory usage is:

$$M_{\text{infer}} = 4 \cdot BS \cdot (N_O + 1) \cdot N_O \tag{70}$$

# D  ADDITIONAL EXPERIMENTS

## D.1  HEURISTIC ANALYSIS OF CONSTRAINED LEARNING AND HYPERPARAMETERS

**Illustrative function fitting**  We aim to fit the illustrative oscillating function $y : \mathbb{R} \to \mathbb{R}^2$, defined by $y_1(x) = 2\sin(fx)$ and $y_2(x) = -\sin^2(fx) - x^2$, where $x \in \mathbb{R}$ is the unidimensional indepen-

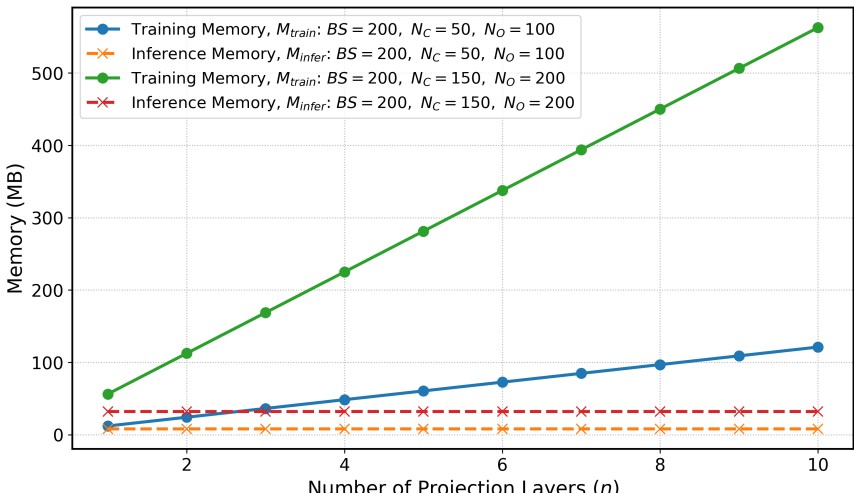

Figure 4: Memory usage during training and inference as a function of the number of projection layers $n$, for two problem scales. $N_C$ is the number of constraints, and $N_O$ is the dimension of the neural network output. Results are shown for a fixed batch size $BS = 200$. Smaller-scale setup: $N_C = 50$, $N_O = 100$; larger-scale setup: $N_C = 150$, $N_O = 200$. Training memory increases linearly with $n$, with significantly higher usage in larger-scale problems. Inference memory remains nearly constant across all configurations.

dent variable (input) and the scalar $f$ is the frequency. Notably, the system is implicitly linked by a nonlinear constraint $c(x, y_1, y_2) = (0.5y_1)^2 + x^2 + y_2$, involving both input and output variables. We train an ENFORCE model consisting of a 64-neuron 1-hidden-layer fully connected ReLU neural network as a backbone and an AdaNP module to force the predictions to satisfy the constraint. The supervised task loss is the mean squared error ($\ell_T = \frac{1}{N} \sum_{i=1}^{N} ||y_i - \tilde{y}_i||^2$), while $\lambda_C$ is set to zero (i.e., the constraint is addressed exclusively by AdaNP and no soft constraint term is used). To verify the regression capabilities, we sample 100 training data points from a uniform distribution in $x = [-2, 2]$ and 100,000 test points in the same domain. Every run is repeated 5 times using different initialization seeds. We compare the method with a traditional (unconstrained) multilayer perceptron (MLP) and a soft-constrained neural network sharing the same architecture. We train all the NNs for 50,000 epochs, using Adam optimizer and a learning rate of $10^{-3}$.

Table 4: Regression accuracy and constraint guarantee of ENFORCE on 100,000 test samples when compared with a multilayer perceptron (MLP) and a soft-constrained neural network (Soft). Results for $\lambda_D = 0.5$, $\epsilon_T = 10^{-4}$, and $\epsilon_I = 10^{-6}$ are reported. We report the inference time for a batch of 1,000 samples, with $f = 5$. (Note that MAPE $= \frac{100\%}{N} \sum_{i=1}^{N} \left| \frac{y_i^* - \tilde{y}_i}{y_i^*} \right|$).

| Method | MAPE [%] | $R^2$ | Mean eq. [%] | Max eq. [%] | Inference [s] |
|---|---|---|---|---|---|
| MLP | $0.339 \pm 0.083$ | $0.994 \pm 0.003$ | $1.47 \pm 0.33$ | $17.13 \pm 3.94$ | $0.002 \pm 0.000$ |
| Soft ($\lambda_C = 1$) | $0.944 \pm 0.143$ | $0.972 \pm 0.002$ | $1.55 \pm 0.16$ | $7.77 \pm 0.40$ | $0.002 \pm 0.000$ |
| ENFORCE | $0.060 \pm 0.028$ | $0.999 \pm 0.000$ | $0.00 \pm 0.00$ | $0.00 \pm 0.00$ | $0.008 \pm 0.003$ |

The main results are summarized in Table 4. ENFORCE outperforms the soft-constrained neural network and the MLP by effectively minimizing the nonlinear constraint residual, guaranteeing arbitrary satisfaction with minor computational costs (Fig. 5c, Appendix D). The inference time for a batch of 1,000 samples is 6 ms longer when using ENFORCE compared to an MLP. This amount should be regarded as additive (+6 ms), not multiplicative (e.g., 4x relative to the MLP). Indeed, the computational complexity is entirely attributed to the AdaNP module, meaning that the backbone architecture has no impact. Therefore, if applied to larger backbones (e.g., transformers), the relative computational impact may become negligible.

We report in the following additional heuristic observations and implications of our constrained

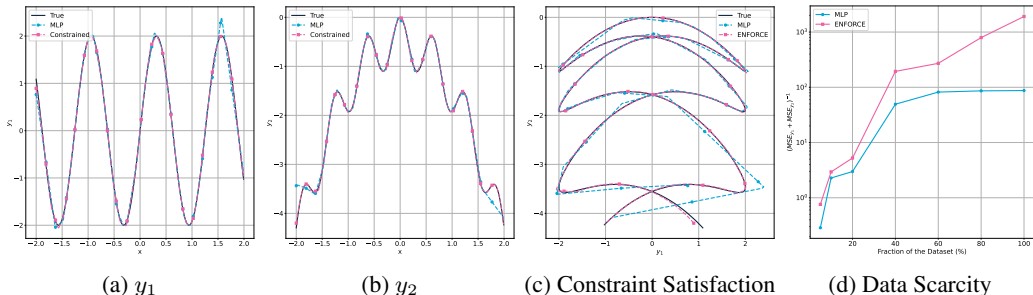

(a) $y_1$      (b) $y_2$      (c) Constraint Satisfaction      (d) Data Scarcity

Figure 5: Prediction comparison between ENFORCE ($\lambda_D = 0.5$, $\epsilon_T = 10^{-4}$, $\epsilon_I = 10^{-6}$) and a multilayer perceptron (MLP). ENFORCE enhances the overall accuracy and guarantees satisfaction for highly nonlinear constraints. ENFORCE consistently performs better than a standard MLP even when trained on uniformly sampled fractions of the training dataset. Interestingly, ENFORCE outperforms the MLP in data-scarce regions of the domain, which in this dataset correspond to the domain extremities (as shown in Fig. 5b). More generally, ENFORCE also performs better under data-scarcity conditions when the models are trained on uniformly sampled fractions of the dataset (Fig. 5d). This observation suggests that constrained learning may enhance data efficiency.

learning routine on the simple function fitting case study. Specifically, we observe (i) the positive influence of our approach on the training dynamics and loss convergence, and (ii) we study the impact of constrained learning hyperparameters such as $\epsilon_T$ and $\lambda_D$.

**Effects of constrained learning** Notably, ENFORCE outperforms the MLP even before the projection steps, demonstrating superior performance using only the neural network backbone (Fig. 6a, dashed-pink line). This can be attributed to the structure of the hard-constrained learning process, where the predictions are adjusted via projection to satisfy underlying constraints. Unlike soft-constrained methods, which only penalize constraint violations in the loss function, hard-constrained optimization incorporates projection-based adjustments that transform predictions to adhere strictly to the constraints. Consequently, after a few training steps, the model benefits from constrained learning, aligning its predictions more closely with valid regions of the solution space, resulting in improved predictions even before projection. Similar insights are also provided by Chen et al. (2021). Therefore, the constrained learning approach is likely to yield improved results even when AdaNP is omitted during inference to enhance computational efficiency. However, it should be noted that in this scenario, constraint satisfaction cannot be guaranteed.

**Training dynamics** To understand the training dynamics of ENFORCE, we analyze the loss curves shown in Fig. 6a, where the training data loss of ENFORCE is compared to the MLP. Being interested in the effect of hard-constrained learning and to ease the visualization, we do not report here the loss curve of the soft-constrained neural network. In this case study, AdaNP contributes to the learning process from the very early iterations (Fig. 6b, orange line), suggesting that the projection operations positively guide the optimization process. The combination of approximated feasible predictions and minimization of projection displacement drives the learning process toward more optimal outcomes.

The modified loss function effectively guides the training process toward smaller projection displacements (Fig. 6b, green dashed line). The displacement loss decreases consistently during training due to the influence of the penalty term in the loss function. Moreover, the depth of AdaNP progressively diminishes over training iterations down to ∼1 layer (Fig.6b, orange line), due to (1) improved overall regression accuracy and (2) smaller projection displacement (i.e., a better linear approximation of the constraints). This adaptive behavior optimizes computational resources by adjusting to the required tolerance at each iteration. Furthermore, this decay in AdaNP depth is consistently observed across different training tolerance values, as illustrated in Fig.8a.

**Constrained learning hyperparameters (training)** We systematically analyze the influence of hyperparameters, such as the displacement weighting factor $\lambda_D$ and the tolerance $\epsilon_T$, on the constrained learning process. Fig. 7 shows the influence of the hyperparameters on the accuracy of trained ENFORCE models evaluated on the test set.

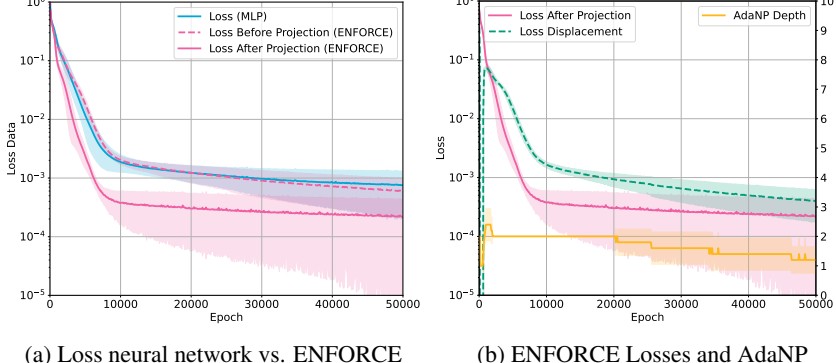

(a) Loss neural network vs. ENFORCE      (b) ENFORCE Losses and AdaNP

Figure 6: ENFORCE demonstrates significantly improved convergence, achieving lower loss compared to an unconstrained MLP. Enhanced training performances are reported for the backbone network of ENFORCE even before the action of AdaNP. This effect is enabled by the simultaneous minimization of the projection displacement (in green) and the action of the AdaNP module (in yellow). Note that we report average values across multiple runs, which explains why the depth of AdaNP appears as a step function with non-integer values.

The impact of the training tolerance $\epsilon_T$ on the model accuracy does not exhibit a clear trend, as its effect varies unpredictably with the weighting factors. Moreover, its influence is generally small compared to the variance of different training runs (Fig. 7). Intuitively, a smaller tolerance $\epsilon_T$ necessitates deeper AdaNP modules, resulting in higher computational costs due to the increased number of neural projection layers. This effect is visible in Fig. 8a, where the depth of AdaNP during training is reported (i.e., number of projection layers). The average depth of AdaNP increases to accommodate stricter tolerances. For example, it expands from one to three layers as the tolerance $\epsilon_T$ is tightened from 1 to $10^{-5}$. Notably, in this case study, AdaNP operates with a minimum of one projection layer (i.e., when the tolerance is set to 1) and a maximum of 100. More importantly, the required depth tends to have a slower decay during training, if compared to using less strict tolerances (as visible in Fig. 8a). Larger tolerances result, on average, in shallow AdaNP layers (approximately one layer). This significantly reduces the training time associated with the projection operations. Along with the minor impact on overall accuracy, this observation suggests setting the training tolerance $\epsilon_T$ to less stringent requirements.

The regression accuracy is evidently affected by the choice of the displacement loss weighting factor $\lambda_D$ (Fig. 7). Remarkably, unlike the challenging task of tuning weighting factors in soft-constrained methods (Wang et al., 2020a), the constrained learning approach proposed here positively impacts accuracy regardless of the specific weighting factor chosen (as shown in Fig. 7, the accuracy of ENFORCE is consistently greater than that of a standard MLP). However, an inappropriate choice of this parameter can result in suboptimal outcomes (e.g., when $\lambda_D = 2$ in Fig. 7). Therefore, careful tuning of this hyperparameter is warranted.

**Constrained learning hyperparameters (inference)** During inference, ENFORCE dynamically adapts the depth of AdaNP to ensure an average tolerance below $\epsilon_I = 10^{-6}$ in this case study. The required depth, however, also depends on the training parameters. Fig. 8b illustrates the number of NP layers needed to satisfy the constraint under varying $\lambda_D$ and $\epsilon_T$. The weighting factor is shown to reduce the required number of NP layers by half, with no additional cost during training. This phenomenon can be attributed to the fact that, in the absence of a penalty for projection displacement, the neural network is free to learn a function that, although potentially far from the actual one, results in projections that fall within the vicinity of the ground truth. This approach, however, necessitates multiple projections. In contrast, the penalty term drives the model to learn a function that is sufficiently close to the ground truth, thereby reducing the number of neural projections required. Increasing the value of $\epsilon_T$ impacts (positively) the depth of AdaNP at inference time when the displacement penalty factor is set to be small during training. This finding further supports the recommendation of employing shallow AdaNP modules during training, by relaxing the value of $\epsilon_T$.

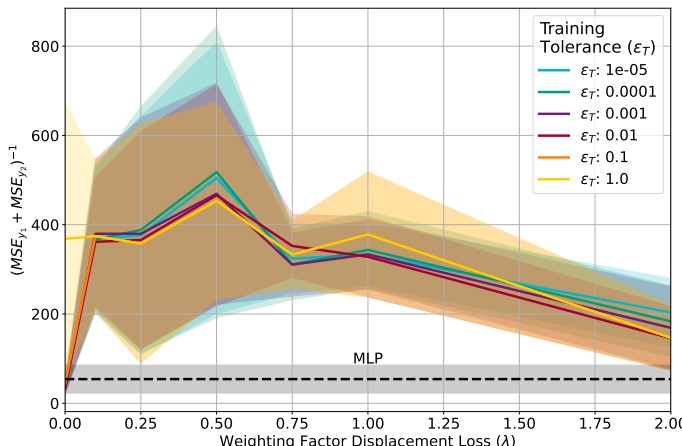

Figure 7: Influence of constrained learning hyperparameters on the accuracy of ENFORCE on the test set (note that here we plot the inverse of the mean squared error (MSE)). The weighting factor $\lambda_D$ favors the learning process if appropriately tuned. Conversely, the training tolerance $\epsilon_T$ exhibits a small impact on performance, suggesting it can be set based on available resources. Overall, despite the choice of hyperparameters, ENFORCE is more accurate than an MLP with the same complexity, while also satisfying the underlying nonlinear constraint.

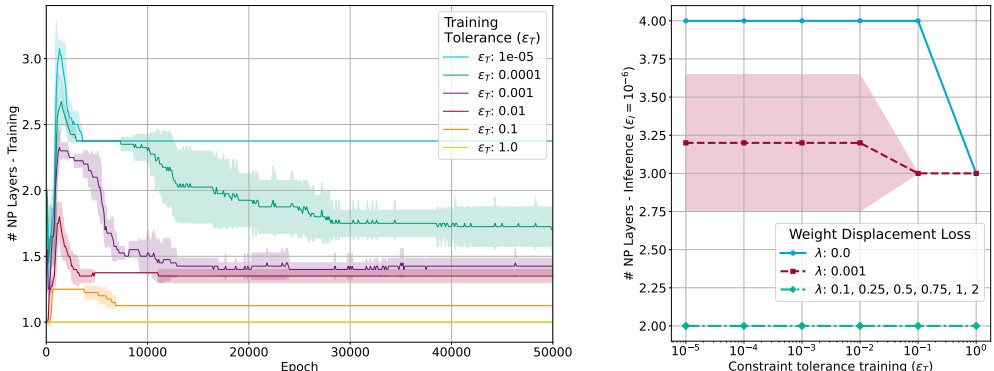

(a) Depth of AdaNP (number of neural projection layers) during training.

(b) Depth of AdaNP (number of neural projection layers) during inference.

Figure 8: Dynamic evolution of AdaNP during training and inference when different training hyperparameters are chosen. At training time, AdaNP is deeper as a smaller constraint tolerance $\epsilon_T$ is chosen.

We conclude that the displacement loss weighting factor $\lambda_D$ plays an important role by balancing the contribution of the projection displacement error. On the other hand, enforcing strict satisfaction during training with an arbitrary small tolerance $\epsilon_T$ does not necessarily improve the overall outcome.

## D.2 Zero-set not locally $C^{1,1}$

We provide an experiment involving a simple regression task with a constraint whose zero set is not locally $C^{1,1}$. Specifically, we consider regressing the functions $y_1 = \sin(x)$ and $y_2 = \sqrt[3]{\sin^2(x)}$, subject to the cusp constraint $c(y_1, y_2) = y_1^2 - y_2^3$. This constraint has a singular point at the origin, where the gradient is non-Lipschitz. We train the model on 100 data points sampled in $[-2, 2]$ and test on 100,000 points. ENFORCE achieves constraint violations below $10^{-6}$ across all test samples, including near the singularity. This result suggests that ENFORCE can handle constraints

that are not locally $C^{1,1}$.

### D.3 SCALING ANALYSIS FOR NONCONVEX PROBLEMS WITH LINEAR AND NONLINEAR CONSTRAINTS

In this section, we present detailed results from the scaling analysis conducted on the two classes of optimization problems presented in Section 5.1: (i) nonconvex problems with linear equality constraints, and (ii) nonconvex problems with nonlinear equality constraints. The following tables report key performance metrics of ENFORCE across varying problem sizes, including different numbers of constraints and optimization variables, and compare them with alternative deep learning-based methods and a traditional large-scale nonlinear programming solver such as IPOPT. Table 5, Table 6 and 7 report the resulting metrics for the linearly constrained case and the nonlinearly constrained case, respectively.

### D.4 DESCRIPTION OF REAL-WORLD CASE STUDY

We adopt a process simulation benchmark reported in prior work (Iftakher et al., 2025), which models the separation of an azeotropic refrigerant using extractive distillation with an ionic liquid. The underlying simulation environment incorporates equilibrium calculations with strong nonlinearities, making the problem computationally demanding and representative of real-world scientific modeling challenges.

The system is governed by mass and energy balance relationships that combine both linear and nonlinear dependencies across inputs and outputs, expressed as physical constraints. The setup defines three controllable input variables. The input space is sampled uniformly over predefined ranges, and for each configuration, steady-state outputs are generated from a simulator. The outputs include multiple flow rates and component fractions, forming a structured multivariate response. For additional details, the reader is referred to Iftakher et al. (2025), Section 3.4.

The study aims to develop a neural network serving as a computationally cheaper surrogate model, while strictly respecting the underlying nonlinear physics of the system.

## E  OUTLOOK AND LIMITATIONS

This work opens several research avenues toward developing robust NNs that strictly adhere to underlying system knowledge. First, the current method can be extended to handle piecewise-defined constraints and nonlinear inequality constraints. Additionally, the requirement for the constraint to be a $\mathcal{C}^1$ function could be relaxed by leveraging sub-gradients (Boyd et al., 2003). Finally, alternative (e.g., weighted) projection approaches could be explored to better account for the morphology and scaling of the constraints.
The method has the potential to address specific challenges or complement existing approaches, including those based on NNs and other machine learning models (e.g., Gaussian processes and support vector machines). For instance, hard constraints can be combined with soft-constraint techniques, such as PINNs, to reliably solve PDEs (Lu et al., 2021). Additionally, the method could enhance learning performance in partially annotated datasets by inferring missing information through available constraints. Finally, an interesting future direction could involve applying the AdaNP module to GenAI models, guiding the generation process toward domain-compliant samples, such as for synthetic data or image/video generation.

**Limitations**   In this paragraph, we highlight the main limitation of the proposed method. Firstly, the effectiveness of ENFORCE is highly dependent on the regression capabilities of the neural network backbone. When the model lacks sufficient complexity to achieve accurate predictions, ENFORCE provides limited benefit. This is supported by the theoretical implications of the orthogonal projections on the constraints manifold discussed in this study. Furthermore, the method becomes computationally and memory-intensive when applied to systems with a large number of constraints (e.g., more than a few hundred). This is due to the computational cost of each neural projection layer, which scales as $O(N_C^3)$ because of the matrix inversion operation, where $N_C$ is the number of constraints. Further details are provided in Appendix C.

Table 5: Scaling experiments on a nonconvex optimization problem with linear equality constraints (Eq. 5) evaluating performance across varying numbers of constraints ($N_C$) and variables ($N_O$). ENFORCE consistently predicts feasible and near-optimal solutions, outperforming alternative deep learning-based methods. DC3 is trained for a greater number of epochs than the other methods, until convergence is reached.

| Constraints | ($N_C$) | 50 | 70 | 150 |
|---|---|---|---|---|
| Variables | ($N_O$) | 100 | 100 | 200 |
| Method | Metric | | | |
| IPOPT | Obj. value | $-11.11 \pm 0.00$ | $-4.84 \pm 0.00$ | $-10.64 \pm 0.00$ |
| | Max eq. | $0.00 \pm 0.00$ | $0.00 \pm 0.00$ | $0.00 \pm 0.00$ |
| | Mean eq. | $0.00 \pm 0.00$ | $0.00 \pm 0.00$ | $0.00 \pm 0.00$ |
| | Inference [s] | $0.095 \pm 0.033$ | $0.13 \pm 0.04$ | $0.379 \pm 0.060$ |
| | Training [min] | – | – | – |
| | Epochs | – | – | – |
| MLP | Obj. value | $-27.43 \pm 0.00$ | $-27.43 \pm 0.00$ | $-52.99 \pm 0.01$ |
| | Max eq. | $24.65 \pm 0.08$ | $24.89 \pm 0.12$ | $45.38 \pm 0.56$ |
| | Mean eq. | $7.32 \pm 0.00$ | $7.19 \pm 0.00$ | $9.14 \pm 0.02$ |
| | Inference [s] | $0.001 \pm 0.000$ | $0.001 \pm 0.001$ | $0.001 \pm 0.001$ |
| | Training [min] | $8.87 \pm 0.18$ | $8.89 \pm 0.11$ | $9.01 \pm 0.25$ |
| | Epochs | 1000 | 1000 | 1000 |
| Soft | Obj. value | $-10.10 \pm 0.31$ | $-1.86 \pm 0.17$ | $1.28 \pm 0.32$ |
| ($\lambda_c = 5$) | Max eq. | $0.53 \pm 0.04$ | $0.79 \pm 0.08$ | $1.45 \pm 0.43$ |
| | Mean eq. | $0.03 \pm 0.00$ | $0.06 \pm 0.00$ | $0.08 \pm 0.00$ |
| | Inference [s] | $0.002 \pm 0.000$ | $0.001 \pm 0.000$ | $0.001 \pm 0.000$ |
| | Training [min] | $10.69 \pm 0.51$ | $10.72 \pm 0.43$ | $10.91 \pm 0.46$ |
| | Epochs | 1000 | 1000 | 1000 |
| Soft | Obj. value | $-10.69 \pm 0.01$ | $-4.18 \pm 0.03$ | $-8.18 \pm 0.18$ |
| ($\lambda_c = 1$) | Max eq. | $0.54 \pm 0.05$ | $0.86 \pm 0.05$ | $1.47 \pm 0.41$ |
| | Mean eq. | $0.05 \pm 0.00$ | $0.08 \pm 0.00$ | $0.09 \pm 0.00$ |
| | Inference [s] | $0.001 \pm 0.000$ | $0.001 \pm 0.000$ | $0.001 \pm 0.001$ |
| | Training [min] | $10.69 \pm 0.52$ | $10.70 \pm 0.46$ | $10.85 \pm 0.49$ |
| | Epochs | 1000 | 1000 | 1000 |
| Soft | Obj. value | $-12.05 \pm 0.00$ | $-6.82 \pm 0.01$ | $-13.55 \pm 0.02$ |
| ($\lambda_c = 0.1$) | Max eq. | $2.09 \pm 0.03$ | $2.51 \pm 0.08$ | $2.17 \pm 0.12$ |
| | Mean eq. | $0.36 \pm 0.00$ | $0.43 \pm 0.00$ | $0.35 \pm 0.00$ |
| | Inference [s] | $0.001 \pm 0.001$ | $0.001 \pm 0.000$ | $0.001 \pm 0.001$ |
| | Training [min] | $10.62 \pm 0.61$ | $10.73 \pm 0.52$ | $10.88 \pm 0.54$ |
| | Epochs | 1000 | 1000 | 1000 |
| DC3 | Obj. value | $-10.31 \pm 10.07$ | $-2.76 \pm 0.06$ | $-6.27 \pm 0.07$ |
| | Max eq. | $0.00 \pm 0.00$ | $0.00 \pm 0.00$ | $0.00 \pm 0.00$ |
| | Mean eq. | $0.00 \pm 0.00$ | $0.00 \pm 0.00$ | $0.00 \pm 0.00$ |
| | Inference [s] | $0.003 \pm 0.000$ | $0.002 \pm 0.000$ | $0.004 \pm 0.000$ |
| | Training [min] | $22.96 \pm 3.73$ | $20.57 \pm 8.30$ | $25.18 \pm 8.63$ |
| | Epochs | 3500 | 3500 | 3500 |
| **ENFORCE** | Obj. value | $-11.50 \pm 0.01$ | $-4.86 \pm 0.00$ | $-10.59 \pm 0.00$ |
| | Max eq. | $0.00 \pm 0.00$ | $0.00 \pm 0.00$ | $0.00 \pm 0.00$ |
| | Mean eq. | $0.00 \pm 0.00$ | $0.00 \pm 0.00$ | $0.00 \pm 0.00$ |
| | Inference [s] | $0.008 \pm 0.001$ | $0.010 \pm 0.001$ | $0.016 \pm 0.002$ |
| | Training [min] | $12.79 \pm 0.03$ | $12.72 \pm 0.04$ | $13.91 \pm 0.07$ |
| | Epochs | 1000 | 1000 | 1000 |

Table 6: Scaling experiments on a nonconvex optimization problem with nonlinear equality constraints (Eq. 6) evaluating performance across varying numbers of constraints ($N_C = [10, 30, 50]$) and variables ($N_O = 100$). ENFORCE consistently predicts feasible and near-optimal solutions. In the simplest setting, it outperforms the nonlinear programming solver IPOPT in terms of solution quality.

| Constraints | $(N_C)$ | 10 | 30 | 50 |
| Variables | $(N_O)$ | 100 | 100 | 100 |
|---|---|---|---|---|
| Method | Metric | | | |
| IPOPT | Obj. value | $-26.27 \pm 0.00$ | $-21.81 \pm 0.00$ | $-18.05 \pm 0.00$ |
| | Max Eq. | $0.00 \pm 0.00$ | $0.00 \pm 0.00$ | $0.00 \pm 0.00$ |
| | Mean Eq. | $0.00 \pm 0.00$ | $0.00 \pm 0.00$ | $0.00 \pm 0.00$ |
| | Inference [s] | $0.094 \pm 0.032$ | $0.244 \pm 0.132$ | $0.268 \pm 0.125$ |
| | Training [min] | $-$ | $-$ | $-$ |
| | Epochs | $-$ | $-$ | $-$ |
| MLP | Obj. value | $-27.43 \pm 0.00$ | $-27.43 \pm 0.00$ | $-27.43 \pm 0.00$ |
| | Max eq. | $214.95 \pm 0.10$ | $317.76 \pm 0.06$ | $317.14 \pm 0.01$ |
| | Mean eq. | $59.49 \pm 0.03$ | $72.10 \pm 0.01$ | $69.63 \pm 0.00$ |
| | Inference [s] | $0.001 \pm 0.001$ | $0.001 \pm 0.001$ | $0.001 \pm 0.001$ |
| | Training [min] | $7.1 \pm 0.3$ | $7.8 \pm 0.0$ | $10.1 \pm 3.1$ |
| | Epochs | 1000 | 1000 | 1000 |
| Soft | Obj. value | $462.93 \pm 29.91$ | $> 10^3$ | $> 10^5$ |
| $(\lambda_c = 5)$ | Max eq. | $16.31 \pm 1.00$ | $71.27 \pm 3.58$ | $72.97 \pm 0.74$ |
| | Mean eq. | $2.31 \pm 0.10$ | $16.19 \pm 0.12$ | $16.81 \pm 0.03$ |
| | Inference [s] | $0.001 \pm 0.000$ | $0.001 \pm 0.000$ | $0.001 \pm 0.001$ |
| | Training [min] | $12.4 \pm 0.1$ | $12.6 \pm 0.3$ | $12.8 \pm 0.6$ |
| | Epochs | 1000 | 1000 | 1000 |
| Soft | Obj. value | $61.93 \pm 3.31$ | $> 10^4$ | $> 10^4$ |
| $(\lambda_c = 1)$ | Max eq. | $15.67 \pm 1.43$ | $70.39 \pm 3.46$ | $72.18 \pm 2.21$ |
| | Mean eq. | $2.16 \pm 0.06$ | $16.21 \pm 0.14$ | $16.81 \pm 0.01$ |
| | Inference [s] | $0.001 \pm 0.000$ | $0.002 \pm 0.000$ | $0.002 \pm 0.001$ |
| | Training [min] | $12.5 \pm 0.1$ | $12.6 \pm 0.3$ | $11.0 \pm 0.6$ |
| | Epochs | 1000 | 1000 | 1000 |
| Soft | Obj. value | $-18.29 \pm 1.21$ | $> 10^3$ | $> 10^3$ |
| $(\lambda_c = 0.1)$ | Max eq. | $16.18 \pm 0.55$ | $70.37 \pm 5.08$ | $76.56 \pm 2.35$ |
| | Mean eq. | $2.05 \pm 0.17$ | $16.35 \pm 0.08$ | $16.86 \pm 0.05$ |
| | Inference [s] | $0.001 \pm 0.000$ | $0.001 \pm 0.000$ | $0.002 \pm 0.001$ |
| | Training [min] | $11.5 \pm 0.1$ | $12.5 \pm 0.4$ | $11.9 \pm 0.1$ |
| | Epochs | 1000 | 1000 | 1000 |
| **ENFORCE** | Obj. value | $-26.37 \pm 0.00$ | $-21.48 \pm 0.01$ | $-16.68 \pm 0.01$ |
| | Max eq. | $0.00 \pm 0.00$ | $0.00 \pm 0.00$ | $0.00 \pm 0.00$ |
| | Mean eq. | $0.00 \pm 0.00$ | $0.00 \pm 0.00$ | $0.00 \pm 0.00$ |
| | Inference [s] | $0.013 \pm 0.002$ | $0.023 \pm 0.002$ | $0.030 \pm 0.005$ |
| | Training [min] | $25.3 \pm 0.1$ | $29.4 \pm 0.1$ | $35.8 \pm 0.4$ |
| | Epochs | 1000 | 1000 | 1000 |

Table 7: Scaling experiments on a nonconvex optimization problem with nonlinear equality constraints (Eq. 6) evaluating performance across varying numbers of constraints ($N_C = [70, 100]$) and variables ($N_O = [100, 200]$). ENFORCE consistently predicts feasible and near-optimal solutions. In the simplest setting, it outperforms the nonlinear programming solver IPOPT in terms of solution quality.

| Constraints
Variables | $(N_C)$
$(N_O)$ | 70
100 | 150
200 |
|---|---|---|---|
| Method | Metric | | |
| IPOPT | Obj. value | $-11.69 \pm 0.00$ | $-29.45 \pm 0.00$ |
| | Max Eq. | $0.00 \pm 0.00$ | $0.00 \pm 0.00$ |
| | Mean Eq. | $0.00 \pm 0.00$ | $0.00 \pm 0.00$ |
| | Inference [s] | $0.401 \pm 0.167$ | $3.40 \pm 1.40$ |
| | Training [min] | $-$ | $-$ |
| | Epochs | $-$ | $-$ |
| MLP | Obj. value | $-27.43 \pm 0.00$ | $-53.07 \pm 0.00$ |
| | Max eq. | $317.97 \pm 0.73$ | $497.38 \pm 4.64$ |
| | Mean eq. | $70.57 \pm 0.01$ | $118.39 \pm 0.07$ |
| | Inference [s] | $0.001 \pm 0.001$ | $0.002 \pm 0.001$ |
| | Training [min] | $9.4 \pm 0.3$ | $10.6 \pm 0.3$ |
| | Epochs | $1000$ | $1000$ |
| Soft
$(\lambda_c = 5)$ | Obj. value | $> 10^5$ | $> 10^5$ |
| | Max eq. | $73.63 \pm 4.56$ | $79.23 \pm 3.81$ |
| | Mean eq. | $16.71 \pm 0.03$ | $16.72 \pm 0.07$ |
| | Inference [s] | $0.002 \pm 0.001$ | $0.001 \pm 0.001$ |
| | Training [min] | $14.0 \pm 0.6$ | $15.1 \pm 0.7$ |
| | Epochs | $1000$ | $1000$ |
| Soft
$(\lambda_c = 1)$ | Obj. value | $> 10^4$ | $> 10^4$ |
| | Max eq. | $73.80 \pm 5.78$ | $79.30 \pm 3.70$ |
| | Mean eq. | $16.71 \pm 0.04$ | $16.68 \pm 0.06$ |
| | Inference [s] | $0.001 \pm 0.001$ | $0.001 \pm 0.000$ |
| | Training [min] | $13.9 \pm 0.6$ | $15.0 \pm 0.6$ |
| | Epochs | $1000$ | $1000$ |
| Soft
$(\lambda_c = 0.1)$ | Obj. value | $> 10^3$ | $> 10^3$ |
| | Max eq. | $75.36 \pm 2.70$ | $78.35 \pm 1.95$ |
| | Mean eq. | $16.77 \pm 0.07$ | $16.63 \pm 0.01$ |
| | Inference [s] | $0.001 \pm 0.001$ | $0.002 \pm 0.001$ |
| | Training [min] | $13.9 \pm 0.7$ | $14.9 \pm 0.8$ |
| | Epochs | $1000$ | $1000$ |
| **ENFORCE** | Obj. value | $-7.75 \pm 0.03$ | $-27.77 \pm 0.02$ |
| | Max eq. | $0.00 \pm 0.00$ | $0.00 \pm 0.00$ |
| | Mean eq. | $0.00 \pm 0.00$ | $0.00 \pm 0.00$ |
| | Inference [s] | $0.049 \pm 0.009$ | $0.14 \pm 0.08$ |
| | Training [min] | $49.0 \pm 0.9$ | $69.4 \pm 23.1$ |
| | Epochs | $1000$ | $1000$ |

