# OpenReview forum: "ENFORCE: Nonlinear Constrained Learning with Adaptive-depth Neural Projection"
_ICLR.cc/2026/Conference — ICLR 2026 Conference Withdrawn Submission_

### Official Review · Reviewer_1MK7 · 2025-10-18

**Soundness:** 2
**Presentation:** 2
**Contribution:** 2
**Rating:** 2
**Confidence:** 5

**Summary:**

This paper proposes ENFORCE, a method to ensure that neural network predictions satisfy a set of nonlinear constraints c(x,y) = 0, without relying on external solvers or incurring significant computational overhead.

**Strengths:**

The proposed approach aims to guarantee strict compliance with nonlinear equalities. The paper presents some theoretical results.

**Weaknesses:**

1. The theoretical results are derived under Assumption 1. However, Assumption 1 is too strong and not practical. Usually, nonlinear equalities have dependence on each other, such as $\dot{p} = v$ and $\dot{v} = f(p)$ for describing a system dynamics or physics law.

2. Equation (2) is used to approximate nonlinear constraints, which will result in approximation error. Due to this approximation error, the proposed approach cannot guarantee the strict satisfaction of the nonlinear constraints, which contradicts the claim of satisfaction.

3. The abstract mentions "Ensuring neural networks adhere to domain-specific constraints is crucial for addressing safety and ethical concerns while also enhancing inference accuracy." Usually, safety and ethical concerns cannot be described by c(x,y) = 0 only. Nonlinear inequalities, i.e., c(x,y) ≤ 0, rather than equalities, can describe them.

4. Recent efforts, see e.g., references R[1] and R[2], have been devoted to guaranteeing the strict satisfaction of nonlinear constraints and outperform the proposed approach. These approaches are the most relevant ones that the paper should compare against. However, the paper presents comparisons with only the most standard MLP. Meanwhile, the paper lacks a corresponding literature review.

5. Depending on the rank condition, the nonlinear equality c(x,y) = 0 can have multiple rather than unique solutions. How can you guarantee the proposed NN framework maintains these properties of the nonlinear equality?


References:

R[1] Chen, Y., Huang, D., Zhang, D., Zeng, J., Wang, N., Zhang, H., & Yan, J. (2021). Theory-guided hard constraint projection (HCP): A knowledge-based data-driven scientific machine learning method. Journal of Computational Physics, 445, 110624.

R[2] Mao, Y., Gu, Y., Sha, L., Shao, H., Wang, Q., & Abdelzaher, T. (2023). Phy-Taylor: Partially Physics-Knowledge-Enhanced Deep Neural Networks via NN Editing. IEEE Transactions on Neural Networks and Learning Systems.

**Questions:**

See Weaknesses.

---

### Official Review · Reviewer_ToJN · 2025-10-24

**Soundness:** 2
**Presentation:** 2
**Contribution:** 2
**Rating:** 2
**Confidence:** 4

**Summary:**

The paper proposes a method for training neural networks whose outputs must satisfy (possibly nonlinear) equality constraints. It introduces an output-projection layer that, at each forward pass, linearizes the constraints around the current prediction, applies a closed-form KKT projection to move toward feasibility, and iterates this projection adaptively until reaching a feasibility tolerance or a depth cap. Theoretical results include existence and uniqueness of a projection within a neighborhood of the feasible set, and local convergence to a feasible output under smoothness/regularity, and non-expansiveness of the projection in the affine case. Experiments on synthetic constrained optimization (linear and nonlinear equalities) and a chemical process prediction task report perfectly feasible outputs and competitive objective values and runtimes relative to differentiable-optimization and penalty baselines.

**Strengths:**

**Originality**. Combines iterative projection over re-linearized constraints with adaptive depth; includes an interesting chemical-process application.

**Quality**.  The authors provide local convergence guarantees for their method. The experiments are reasonably controlled across linear and nonlinear settings, reporting feasibility, runtime, and task performance.

**Clarity**. The proposed method is fairly easy to understand. Algorithmic choices and implementation details are well documented.

**Significance**. Feasibility at inference is valuable in many safety-critical domains; the layer is optimizer-agnostic and likely easy to integrate across applications.

**Weaknesses:**

### Major Weaknesses: Novelty and Core Claims

1. **Misleading Theoretical Guarantees**: The main text overgeneralizes key results—specifically the 1-Lipschitz stability (Theorem 2) and the “no-worse” property (Proposition 2)—as if they hold for the full nonlinear setting. In reality, the proofs in the appendix rely on convexity or affine-set assumptions, meaning these guarantees apply only to linear constraints. This distinction is critical and should be clearly stated in the main text, not confined to appendix assumptions.

2. **Discrepancy in Feasibility Claims**: The method is framed as enforcing constraints “by construction,” implying a strict, hard guarantee. In practice, however, it relies on an iterative projection process that achieves only feasibility up to a tolerance, and even that is proved only locally, requiring predictions to start sufficiently close to the constraint manifold and and may terminate at an iteration cap with nonzero residual. This gap between the paper’s hard-constraint framing and its conditional, approximate guarantees should be clearly acknowledged in the main text.

3. **Missing Universal Approximation Theory**: The paper does not analyze if the proposed projected layer limits the network's approximation ability within the feasible space, unlike other constrained methods (e.g., HardNet). The authors should either prove that the proposed method preserves universal approximation, or discuss limitations and contrast with HardNet’s guarantee.

### Significant Experimental Weaknesses

4. **Missing Comparison to HardNet**. Table 1, which involves a problem with linear constraints, should include HardNet as a baseline.

5. **DC3 Omission**: Please clarify why DC3 was not included in Table 2. DC3 supports nonlinear constraints and scales to ~100 constraints (see Table 2 in the DC3 paper), comparable to your 150-constraint setup. Was it inapplicable, or simply omitted?

6. **Poor Soft Baseline**: The soft-constrained method performed exceptionally poorly, suggesting inadequate penalty coefficient tuning. Stronger baselines using better tuned or adaptive penalties (e.g., Lagrangian methods) should have been considered.

7. **Missing Initialization Sensitivity**: Since convergence is only local, the method is likely sensitive to initialization, but no ablation was provided.

### Minor Issues

8. **Computational Cost Misrepresented**: Each projection step involves an $O(N_C^3)$ matrix inversion, but this operation is repeated for $n$ projection layers, yielding an actual per-batch runtime of $O(n \cdot N_C^3)$. Moreover, the claimed ``batch-size independence'' arises from GPU parallelization of the inversions rather than true algorithmic scaling, and it overlooks the *per-sample* Jacobian computations that still scale with batch size.. Because both feasibility and runtime depend strongly on the adaptive depth $n$, the paper should report depth statistics (e.g., mean, max, and percentage hitting the cap) along with corresponding feasibility outcomes.

9. **Contradictory Stability Argument**: The paper criticizes null-space methods like DC3 for potential instability, noting that inverting a block of the constraint Jacobian may be ill-conditioned. However, the proposed method itself relies on computing the inverse of $BB^\top$, where $B$ is the constraint Jacobian at each layer. Consequently, it is subject to the same numerical conditioning issues whenever the constraint Jacobian $B$ is poorly conditioned. This should be explicitly acknowledged by the authors.

10. **Unacknowledged Gradient Approximation as a Limitation**: The approximate way of computing gradients–not backpropagating through the constraint Jacobian–should be explicitly acknowledged as a potential limitation of the method. While ignoring these terms is reasonable for computational efficiency, the paper does not discuss its possible negative implications for training dynamics, convergence behavior, or final model performance. A design choice of this nature also warrants an ablation to justify the computational gains relative to the (presumably small) accuracy trade-off.

11. **Lack of Polish and Clarity**: The paper's presentation hinders readability. The main text is verbose yet vague, relegating essential details—including core algorithms, precise theorem statements, and justifications for design choices—to the appendix. This forces the reader to constantly consult the appendix to understand the method. Furthermore, the writing is often imprecise, with unclear notation (e.g., omitting the network's role in constraints) and loosely formulated theoretical statements (e.g., Prop. 1). Necessary context, like the mathematical details of the chemical benchmark experiments, is also missing, requiring to check external references.

**Questions:**

1. The convergence proof provides only local feasibility guarantees and makes no claims about optimality. Do the authors have any concrete results or analysis regarding suboptimality of their method?


2. Several nonlinear experiments report a constraint violation of exactly 0, with zero variance, implying perfect feasibility across all runs. However, since the projection layer enforces only linearized constraints and convergence is local and only required up to a set tolerance, such results appear implausibly optimistic. Can the authors confirm whether these reported zeros are due to rounding or averaging below tolerance rather than truly exact feasibility?

---

### Official Review · Reviewer_hVez · 2025-10-24

**Soundness:** 3
**Presentation:** 4
**Contribution:** 3
**Rating:** 6
**Confidence:** 4

**Summary:**

This paper presents a method for enforcing nonlinear equality constraints on the output of neural networks. The method works by adding a series of differentiable neural projection layers (onto any standard backbone) that project the output onto a linearized version of the constraints. The authors propose a method to adapt the depth of these layers during training and inference to achieve a specified tolerance similar to sequential quadratic programming. They test their method on a proposed optimization benchmark and a neural network surrogate of a chemical process. The method outperforms the baseline in terms of accuracy and inference time.

**Strengths:**

- The paper is well-written and provides an extensive review of related work.
- The proposed method is fully differentiable and can be added to any standard neural network backbone.
- The adaptive depth is a clever feature that can be adjusted depending on the required precision.
- The method outperforms baselines with soft constraints in terms of accuracy and outperforms the baseline that enforces hard constraints.

**Weaknesses:**

- It is not clear how many different types of problems for which the proposed method is useful since only one real-world benchmark is provided.
- The authors note in the introduction that incorporating constraints into neural networks may help with regulatory requirements or restrictions on GenAI. However, it is not necessarily straightforward to encode these requirements as nonlinear constraints.
- The networks used in the experiments are small, and it is not clear how the method scales to large problem sizes empirically (the authors do, however, make some remarks on the complexity in terms of number of constraints).

**Questions:**

How does the methodology scale with a high-input or output dimension? It would be useful to get a better understanding of how the method scales to more complex problems both in terms of dimensionality and problem structure (i.e. what are some examples of nonlinear constraints for GenAI or regulatory requirements?).

---

### Official Review · Reviewer_VWwj · 2025-10-30

**Soundness:** 4
**Presentation:** 3
**Contribution:** 4
**Rating:** 8
**Confidence:** 4

**Summary:**

This paper focuses on developing neural network models that strictly enforces constraints/requirements on the prediction, with optimization proxies being the focus of the paper.  There are number of ways the literature has pursued to solve this problem, and this paper focuses on projection methods, where static layers are added to the end of the neural network that project the predicted solution back into the feasible region of the optimization problem (sometimes called a repair layer in the literature).  The drawback of these methods is that the projection operator can introduce computationally complexity that slows down the training iterations dramatically.  This paper seeks to address this problem by recasting the projection as a sequence of layers that essentially mimic the operations of sequential quadratic program (SQP) algorithm.  These layers are well behaved in standard neural network training (easily differentiable) and individual layers do not substantially impact the computation of training step.  The main drawback is that in order to guarantee predictions that always satisfy constraints, the network no longer has a static size, so the approach could introduce a substantial number of layers (each layer corresponding to an iteration of SQP, with number of iterations not known apriori) that also slows down the training.  The authors introduce a regularization term that seems to prevent a large number of layers from being necessary.

**Strengths:**

1.	Very clean approach to model SQP as layers in a neural network to project NN predictions to solutions that lie within a feasible region
2.	Very strong empirical results on a variety of nonlinear problems

**Weaknesses:**

1.	Main weakness is that the approach may need an arbitrarily deep network to get epsilon satisfaction of constraints – in both training and inference.  I view this weakness as somewhat minor given the introduction of a regularization term to control constraint violations and associated empirical evidence.
2.	I feel like some of the core pieces of the paper were relegated to the appendix. Specifically, section C.1 which describes how the adaptive layers are rolled out and the sections on constrained learning hyperparameters, which describe/show how the depth of the AdaNP layers is kept under control.  I realize the authors are space limited, but those two sections were crucial for understanding and evaluating the contribution.

**Questions:**

1.	As I understand the implementation, the depth of the NN (for AdaNP layers) is completely dynamic and is different for every input x. Is this correct? And this can vary for each x at every iteration of the training?

---

### Note · Authors · 2025-12-02

I have read and agree with the venue's withdrawal policy on behalf of myself and my co-authors.